# Two-dimensional mineral hydrogel-derived single atoms-anchored heterostructures for ultrastable hydrogen evolution

Fucong Lyu [1,2], Shanshan Zeng [3], Zhe Jia [2,4], Fei-Xiang Ma [1,2], Ligang Sun [5,6] ✉, Lizi Cheng[1,2], Jie Pan [3], Yan Bao[1,2], Zhengyi Mao [1,6], Yu Bu [1,2], Yang Yang Li [3] ✉ & Jian Lu [1,2,6,7] ✉

Hydrogen energy is critical for achieving carbon neutrality. Heterostructured materials with single metal-atom dispersion are desirable for hydrogen production. However, it remains a great challenge to achieve large-scale fabrication of single atom-anchored heterostructured catalysts with high stability, low cost, and convenience. Here, we report single iron (Fe) atom-dispersed heterostructured Mo-based nanosheets developed from a mineral hydrogel. These rationally designed nanosheets exhibit excellent hydrogen evolution reaction (HER) activity and reliability in alkaline condition, manifesting an overpotential of 38.5 mV at 10 mA cm$^{-2}$, and superior stability without performance deterioration over 600 h at current density up to 200 mA cm$^{-2}$, superior to most previously reported non-noble-metal electrocatalysts. The experimental and density functional theory results reveal that the O-coordinated single Fe atom-dispersed heterostructures greatly facilitated $H_2O$ adsorption and enabled effective adsorbed hydrogen (H*) adsorption/ desorption. The green, scalable production of single-atom-dispersed heterostructured HER electrocatalysts reported here is of great significance in promoting their large-scale implementation.

Hydrogen ($H_2$) is an attractive source of energy due to its high energy density and emission of few pollutants when combusted in air[1–3]. Among the wide range of $H_2$ generation methods, the electrochemical HER is particularly facile and economical[4,5]. The high cost of commercial noble metal-based HER electrocatalysts motivates researchers to develop low-cost high-activity electrocatalysts with high electrical conductivity and high stability on a large scale, which remains a challenging task[6,7]. Single-atom catalysts (SACs) have been extensively explored for use in catalytic HER application due to their high activity,

maximized atomic efficiency and minimized catalyst usage of a given process[8,9]. Many efforts had been devoted to constructing various kinds of single-atom catalysts, such as single-atom/carbon, single-atom/metal oxide, single-atom/metal sulphide, single-atom alloy[10].

Single-atom catalysts were usually fabricated by anchoring targeted single-atom metal on different substrates by introducing the targeted single-atom metal to the substrate precursor and following a post-heating treatment[11,12]. In order to achieve favorable loading of the single atom and strong interaction between single-atoms and the

[1]Centre for Advanced Structural Materials, City University of Hong Kong Shenzhen Research Institute, Greater Bay Joint Division, Shenyang National Laboratory for Materials Science, Shenzhen 518057, China. [2]Hong Kong Branch of National Precious Metals Material Engineering Research Centre, City University of Hong Kong, 83 Tat Chee Avenue, Kowloon, Hong Kong, China. [3]Department of Material Science and Engineering, City University of Hong Kong, 83 Tat Chee Avenue, Kowloon, Hong Kong, China. [4]School of Materials Science and Engineering, Southeast University, Nanjing 211189, China. [5]School of Science, Harbin Institute of Technology, Shenzhen 518055, China. [6]Department of Mechanical Engineering, City University of Hong Kong, 83 Tat Chee Avenue, Kowloon, Hong Kong, China. [7]CityU-Shenzhen Futian Research Institute, Shenzhen 518045, China. ✉e-mail: sunligang@hit.edu.cn; yangli@cityu.edu.hk; jianlu@cityu.edu.hk

substrates, large number of defects, heterogenous atom dopants, favorable facets or abundant interfaces in the substrates are required,which greatly increase their fabrication difficulty[11]. Thus, simplifying the preparing process for their substrates and increasing the universality in dispersing targeted single atoms are vital to expand the diversity and prevailingness of single-atom materials. Nevertheless, most of reported methods suffer from complex fabrication process, low dispersion of single-atom metal, high post-treatment temperature (usually higher than 700 °C), and environmentally unfriendly issues involved, impeding the practical use of SACs. In addition, to maximize the exposure of the single atomic sites, the substrates prefer to low dimensional, porous, and abundant grain boundaries, which brings even higher requirements for SACs preparation. For example, using the porous metal-organic frameworks (MOFs) or covalent organic frameworks (COFs) as a precursor to construct SACs is recently a promising method[13,14]. However, the synthesis and purification of MOFs/COFs, as well as the follow-up metal species absorption process are extremely a time-consuming process. In other practices, the targeted metal species were added in advance during the MOFs/COFs preparation process, which may disturb the frameworks formation of MOFs/COFs. In addition, the utilization of organic, complex synthetic parameters, high and precisely-control of pyrolysis temperature, release of organic exhaust gas during MOFs/COFs preparation and their carbonization process, are regarded as unsustainable, unfriendly, and tedious methods, seriously restricting the flexibility and universality of this strategy[15]. These issues substantially impede the feasibility of this method. Oppositely, the synthesis of heterostructure constructed by metal oxides/phosphides for SACs preparation could be easier and more convenient[16]. Heterostructure featured with the low dimensional and porous characters can vastly increase the loading of single-atoms and the expose of active sites, making it an ideal single-atom carrier. Therefore, it is urgent to develop robust single-atom-dispersed heterostructured nanosheets featuring simplicity, environmentally benignity, and high effiency[17,18].

Mineral hydrogel is made by inorganic matter, inorganic salt and water, and can self-assemble into gel networks with various morphologies via a rapid and straightforward process of mixing and gelation under mild conditions[19,20]. The coordinated metal ions, which are singly dispersed in the gel networks, can convert into a heterostructure of metal oxides/phosphides/sulphides following a post-heating treatment. A single Fe atoms N-doped carbon was successfully prepared with a Fe-Polypyrrole supramolecular hydrogel by Yu's group according to the highly dispersed metal character[21]. However, the conversion of supramolecular hydrogel to a carbon usually need a high temperature and the yield of carbonization is usually low. Encouragingly, some metals can keep single dispersion state in the heterostructured material derived from mineral hydrogel due to the strong interaction with the abundant oxygen/phosphorus atoms and plentiful interfaces, and can convert to heterostructured material at a low temperature. Therefore, mineral hydrogel could act as an excellent candidate for single-atom catalyst synthesis. In addition, the strong ionic character of the mineral hydrogel can easily accommodate highly dispersed metal ionic additives and transition metal salts which means that the targeted single-atom can be easily manipulated. These additives can be added before the gelation process, avoiding the long-time metal absorption procedure of the porous precursor. More importantly, we can expand the metal selection range to cheaper mineral metals to construct mineral hydrogel, alleviating the high cost of precious metals and unfriendly environmental problems during the synthetic process. In summary, compared with other common single atom substrate precursors (porous frameworks and carbon), mineral hydrogels show great advantages in terms of synthetic route, environment-friendly, production efficiency, tunability, raw materials, sustainability, universety, and cost-effectiveness (Fig. 1a and Supplementary Table 1).

In this work, we propose using Mo-based mineral hydrogel as a precursor for designing a novel carbon (C)-free single Fe-atom-dispersed heterostructured nanosheets. Firstly, a novel 2D wrinkle-like iron–phosphomolybdic acid mineral hydrogel nanosheets (FePMoG) was simply synthesized in one step as a precursor, which was directly converted to carbon (C)-free single Fe-atom-dispersed heterostructured Mo-based nanosheets (Fe/SAs@Mo-based-HNSs) via a one-step low-temperature phosphorisation process. The 2D wrinkle-like nanosheet morphology of FePMoG is entirely different and new in mineral hydrogel, and the preparation of FePMoG is simple and environmentally benign through a self-assembly inorganic–inorganic coordination process: that is, in water at room temperature without the addition of any surfactants or additives. The porosity and heterostructure of the phosphorised nanosheets derived from the FePMoG nanosheets were easily regulated during the phosphorisation process by adjusting the reaction temperature and time. It is important to mention that this single-atom-dispersed heterostructured nanosheets was first developed from a mineral hydrogel. We found that Fe/SAs@Mo-based-HNSs exhibited excellent electrocatalytic activity and long-term durability in the HER: an overpotential of 38.5 mV at 10 mA cm$^{-2}$ with a Tafel slope of 35.6 mV dec$^{-1}$ and a negligible increase in polarisation potential and overlapping polarisation curves upon 600 h continuous operation at 200 mA cm$^{-2}$. This is one of the highest performances in the HER demonstrated by any nanostructured Mo-based electrocatalyst, and is attributable to the optimized single atom-dispersed heterostructure of Fe/SAs@Mo-based-HNSs, with a large active surface area and high porosity. This heterostructure contains numerous interfaces and a facile O-coordinated monoatomic dispersed Fe atoms, and thus has high $H_2O$ adsorption ability ad appropriate H* adsorption/desorption ability.

## Results
### Synthesis and characterization of as-prepared FePMoG and Fe/SAs@Mo-based-HNSs eletrocatalyst

Figure 1b illustrates our novel coordination-induced self-assembly process for preparing the FePMoGs nanosheets. This process involved simply mixing a solution of polyoxometallate acid (PMo) and ferric ions (Fe$^{3+}$) at room temperature, which is much more convenient than previously reported processes that typically require a template or solvothermal treatment[22,23]. A suspension of homogeneously distributed FePMoG nanosheets was formed by this process of coordination-induced self-assembly, and was subsequently centrifuged (Centrifugation speeds: 447 g) to remove excess water to enable the next step (Supplementary Fig. 1). During the self-assemble process, the empty orbital in Fe$^{3+}$ species coordinated with the molybdate groups of PMo, resulting in the self-assembly of FePMoG compounds. A 50-fold amplification of the synthesis experiment for this mineral hydrogel was successfully conducted (Supplementary Fig. 2), together with the simple experimental condition and procedure, indicating our method is amenable for large-scale production. Concentration-dependent experiment with various molar ratios of Fe$^{3+}$: PMo shows that the mixture comprising a 25:1 molar ratio of Fe$^{3+}$: PMo can form wrinkled nanosheet (Fig. 1c, Supplementary Fig. 3–5) with a width of micrometres and a thickness of ~50 nm, whose surface was otherwise smooth. Time-dependent experiment of the self-assembly FePMoG nanosheet (25:1) revealed that the wrinkled nanosheets gradually evolved from apophyses (Fig. 1b, Supplementary Fig. 6, 7). To demonstrate the applicability of the mineral hydrogel for producing other monatomic dispersed catalysts, different ions (e.g., Co$^{2+}$, Ni$^{2+}$, Cu$^{2+}$, Ag$^{+}$, or Mn$^{3+}$, ~ 3 at.% of the sum of Fe and Mo atoms) were added to the Fe$^{3+}$ solution and mixed with the phosphomolybdic acid to prepare different mineral hydrogels. As shown in Supplementary Fig. 8, the addition of the other ions did not affect the formation of the mineral hydrogel.

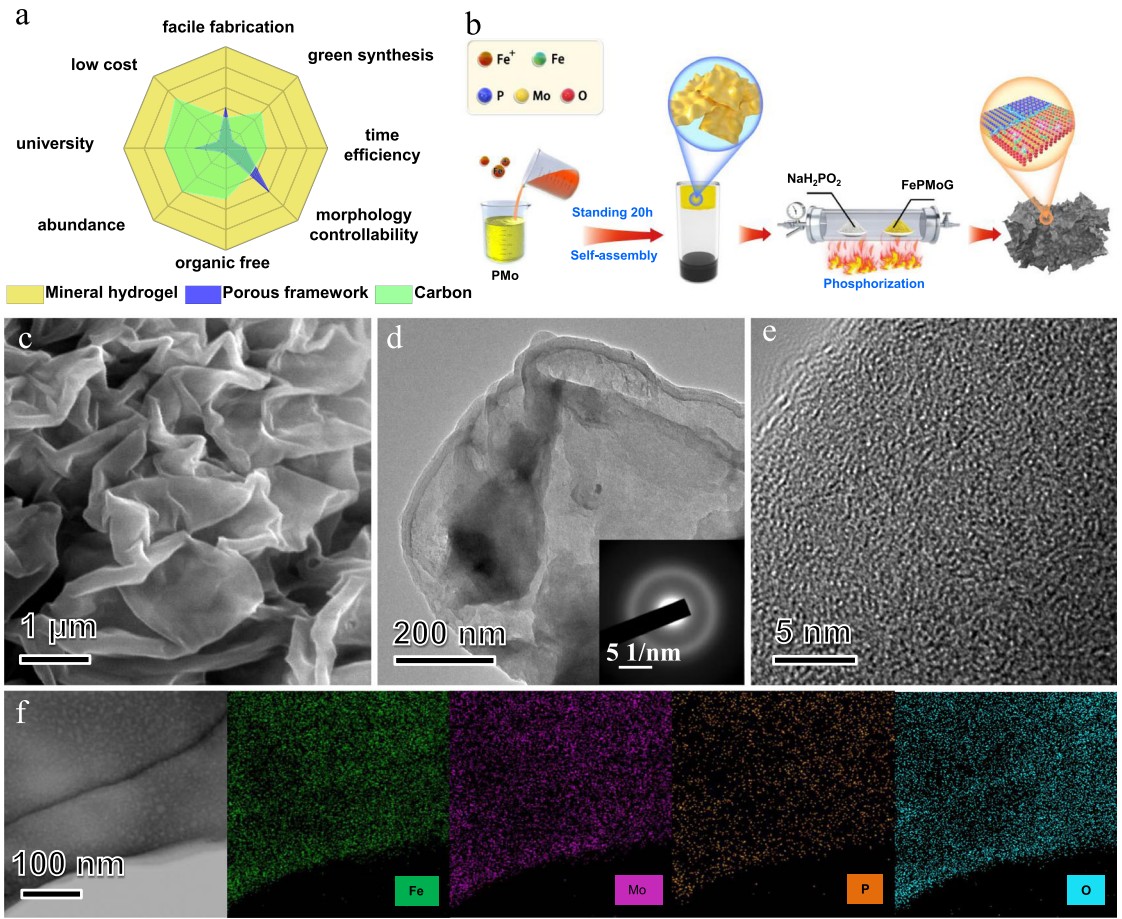

**Fig. 1 | Conceptual design and microstructural characterisation of the as-prepared FePMoG and Fe/SAs@Mo-based-HNSs eletrocatalyst. a** General characteristics comparison of the common substrate precursors for SACs (more details presented in Table S1); **b** schematic of synthesis of the Fe/SAs@Mo-based-HNSs eletrocatalyst; **c** SEM image, **d** TEM image, with inset of the SAED pattern, **e** high-resolution TEM image, and **f** STEM image of an FePMoG with an $Fe^{3+}$: PMo molar ratio of 25:1, and the corresponding EDS elemental mapping of Fe (green), Mo (violet), phosphorus (P; orange), and oxygen (O; cyan).

The X-ray diffraction (XRD) pattern of the FePMoG (25:1) has a prominent broad peak at 27.5° and a weak broad peak at 55° (Supplementary Fig. 9a), which are completely distinct from the sharp peaks of the crystalline structures of pure PMo and iron (III) nitrate nonahydrate[24,25], indicating the amorphous state and lamellar structure of this FePMoG and the strong interactions between the phosphomolybdate and metallic ions[26]. Fourier-transform infrared (FTIR) spectroscopy of FePMoG (25:1) was performed to determine the interactions of its chemical bonds (Supplementary Fig. 9b), which revealed a distinct set of bands characteristic of a Keggin structure derived from PMo[27–29], and strong interactions between Mo and various forms of O as well as between $Fe^{3+}$ and PMo–heteropolyanions[27,28]. The slight shift of the Mo (VI) oxidation state to a higher binding energy (from 232.3 to 232.4 eV for $Mo^{6+}$ $3d_{3/2}$ and from 235.4 to 235.6 eV for $Mo^{6+}$ $3d_{5/2}$ in the Mo 3d X-ray photoelectron spectroscopy (XPS) curve) in the FePMoG is also attributable to the coordination between phosphomolybdate and $Fe^{3+}$ (Supplementary Fig. 10)[30]. These FTIR and XPS data indicate that this FePMoG is formed by strong interactions between $Fe^{3+}$ and PMo.

Transmission electron microscopy (TEM) was used to further analyse the structure and elemental distribution of FePMoG. The TEM image of the FePMoG in Fig. 1d demonstrates the 2D structure of FePMoG composites, and the selected-area electron diffraction (SAED) patterns (inset in Fig. 1d) contain a broad and diffuse ring, confirming this compound's amorphous nature. High-resolution TEM images (Fig. 1e) show the disordered arrangement of atoms in the amorphous FePMoG. The scanning transmission electron microscopy (STEM)

image and corresponding energy-dispersive spectroscopy (EDS) elemental mapping (Fig. 1f) of FePMoG reveal the uniform distribution of Fe, Mo, P, and O atoms within its structure, indicating that it consists of uniform interactions between $Fe^{3+}$ and PMo.

The structural evolution of FePMoG nanosheets after phosphorization is shown in Fig. 2 and Supplementary Fig. 11. FeMoP-T represents samples generated at various pyrolysis temperatures (T); for example, FeMoP-500 represents Fe/SAs@Mo-based-HNSs generated by pyrolysis at 500 °C. XRD patterns of Fe/SAs@Mo-based-HNSs reveal the characteristic diffraction peaks of molybdenum dioxide ($MoO_2$, JCPDS No. 78-1069), molybdenum phosphide (MoP, JCPDS No. 24-0771), and molybdenum diphosphide ($MoP_2$, JCPDS No. 89-2678) (Fig. 2a). We also determined that the heterostructure of the interconnected nanoparticle subunits could be tailored by varying the pyrolysis temperature (Supplementary Fig. 11 and Supplementary Table 2). Iron phosphide (FeP; JCPDS No. 89-2597) and $MoO_2$ phases were formed when the pyrolysis temperature was <450 °C and <550 °C, respectively, whereas MoP and $MoP_2$ phases were formed when the pyrolysis temperature was ≥400 °C and ≥450 °C, respectively. The 2D porous nanosheet structure of the Fe/SAs@Mo-based-HNSs maintained its initial nanosheet morphology during the process of self-templated phosphorisation (Fig. 2b), whereas a simple mixture of an Fe salt and PMo (denoted as bulk FeMoP-500) subjected to phosphorisation produced micrometre-scale aggregates of heterostructured nanoparticles (Supplementary Fig. 12). This indicates that the formation of the strongly coordinated FePMoG plays a decisive role in the phosphorisation-mediated formation of the 2D porous

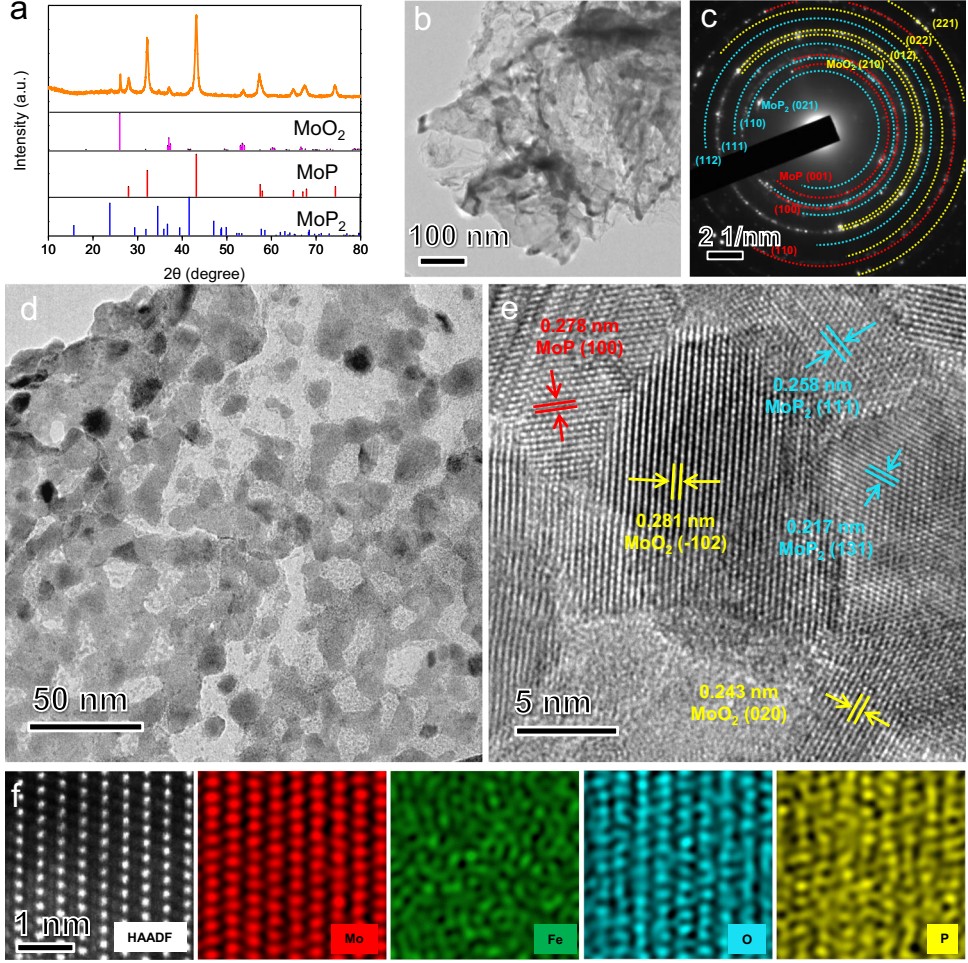

**Fig. 2 | Structural characterisation of Fe/SAs@Mo-based-HNSs. a** XRD patterns; **b** low-magnification TEM image, **c** SAED pattern, **d** high-magnification TEM image, **e** HRTEM image, and **f** HAADF-STEM image and the corresponding EDS elemental mapping of Fe (green), Mo (violet), P (orange) and O (cyan).

nanosheet structure of Fe/SAs@Mo-based-HNSs. The corresponding SAED pattern contains rings ascribable to $MoO_2$, MoP, and $MoP_2$ (Fig. 2c). A high-magnification TEM image shows that Fe/SAs@Mo-based-HNSs (Fig. 2d) comprise an interconnected structure of pores with an average diameter of <10 nm. The lattice fringes in Fig. 2e were found to be 0.281, 0.243, 0.278, 0.209, 0.258, and 0.217 nm wide, corresponding to the interplanar distance of $MoO_2$ (−102), $MoO_2$ (020), MoP (100), MoP (101), $MoP_2$ (111), and $MoP_2$ (131), respectively, indicating the heterostructure of Fe/SAs@Mo-based-HNSs. The HAADF-STEM image of the Fe/SAs@Mo-based-HNSs and the corresponding EDS elemental mapping (Fig. 2f) reveal the uniform distribution of Fe, Mo, P, and O atoms and the Fe atoms were independently dispersed in the lattice, validating that an in-situ transformation occurs during the phosphorisation process and the single atoms dispersion of Fe. The samples formed at other pyrolysis temperatures have similar heterostructures, e.g., FeMoP-450 has four phases (FeP, $MoO_2$, MoP, and $MoP_2$; Supplementary Fig. 13), and FeMoP-550 has three phases ($MoO_2$, MoP, and $MoP_2$; Supplementary Fig. 14), respectively. This demonstrates that the heterostructure in the phosphorised materials were easily obtained and modulated by controlling the annealing temperatures, which is accordance with the above XRD results.

The Brunauer–Emmett–Teller (BET) and pore-size distribution of the Fe/SAs@Mo-based-HNSs show a surface area of 89.6 $m^2\,g^{-1}$ with many mesopores (pore diameters 1.3–12 nm) (Supplementary Fig. 15). This mesoporous and interconnected 2D structure is likely responsible for the efficient electrocatalysis of the HER by Fe/SAs@Mo-based-

HNSs, as this structure would greatly enhance the accessibility of the rich active sites of Fe/SAs@Mo-based-HNSs and increase the infiltration of electrolytes, thereby facilitating charge transfer.

XPS and synchrotron-radiation-based hard X-ray absorption near-edge structure (XANES) were performed to characterise the chemical composition, valence and coordination states, and electronic structure of Fe/SAs@Mo-based-HNSs (Fig. 3). The full-scan XPS spectrum confirms that Fe/SAs@Mo-based-HNSs comprise Fe, Mo, P, and O atoms (Supplementary Fig. 16). As depicted in Fig. 3a, the high-resolution XPS spectrum of Mo 3d consists of six peaks representing three ionic species: 231.66 and 228.34 eV ($Mo^{3+}$), 232.58 and 229.68 eV ($Mo^{4+}$), and 235.63 and 233.49 eV ($Mo^{6+}$), corresponding to MoP, $MoO_2$, and $MoP_2$, respectively[31–33]. The Mo K-edge XANES spectrum shows that the near-edge absorption energy of Fe/SAs@Mo-based-HNSs is intermediate between that of Mo foil and $MoO_2$, demonstrating that the average oxidation state of Mo is between $Mo^0$ and $Mo^{4+}$ (Fig. 3d). The corresponding extended X-ray absorption fine structure (EXAFS) reveals that dispersed Mo atoms in Fe/SAs@Mo-based-HNSs are coordinated with P and O atoms, with no Mo–Mo bonding apparent (Fig. 3e, f, Supplementary Fig. 17, and Supplementary Table 3).

In terms of the oxidation state of Fe, the high-resolution XPS spectrum of Fe (Fig. 3b) has peaks at 710.7 eV (Fe $2p_{3/2}$) and 724.1 eV (Fe $2p_{1/2}$), which are characteristic of $Fe^{3+}$, and an $Fe^{3+}$ 2p satellite peak at 715.6 eV. Similarly, the Fe K-edge XANES spectrum also shows that the valency of Fe is close to that of $Fe^{3+}$ (Fig. 3g). The corresponding EXAFS results delineate the coordination environment of Fe atoms: the strong peak at ~1.5 Å is attributable to Fe–O bonding,

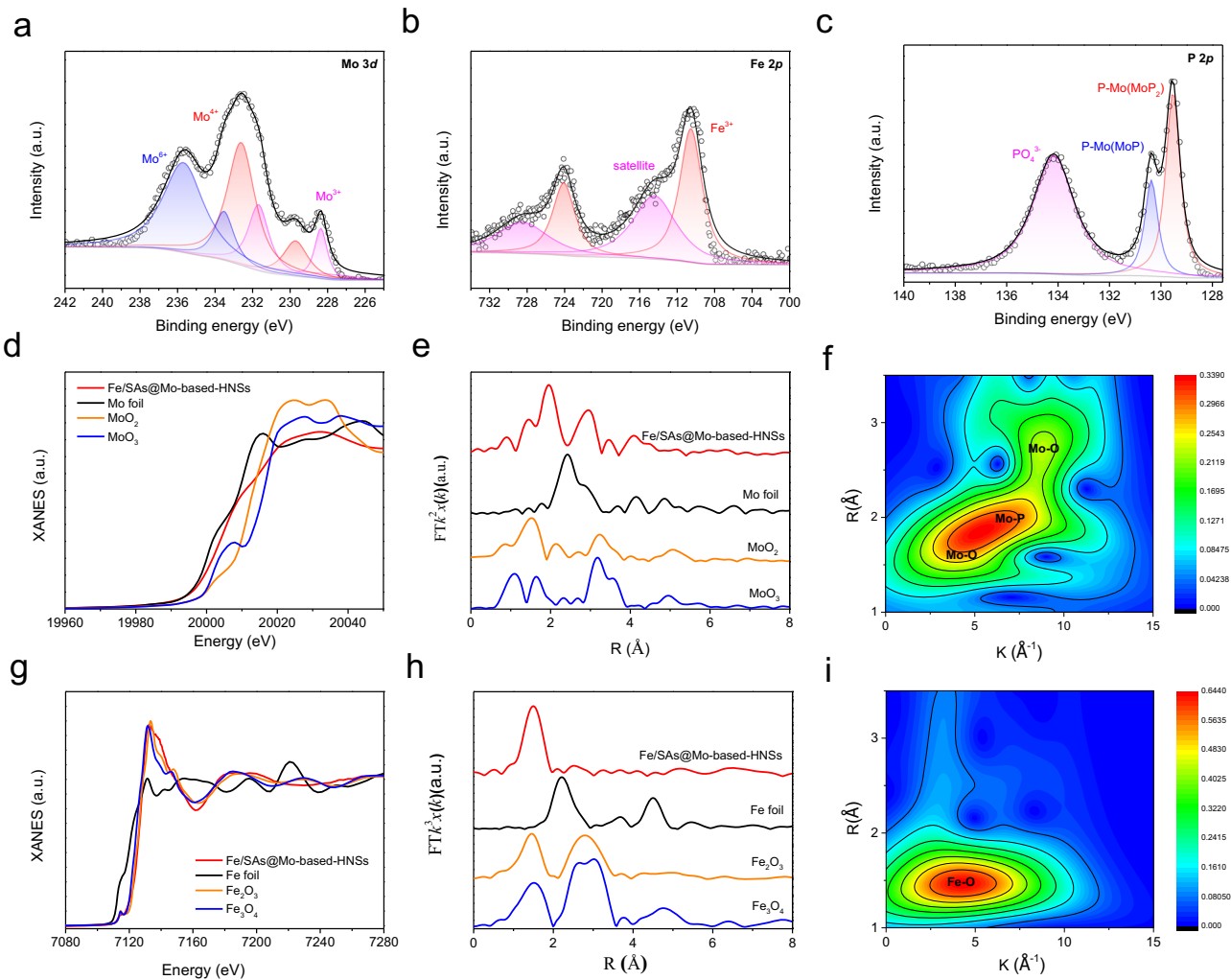

**Fig. 3 | Spectroscopy of Fe/SAs@Mo-based-HNSs. a** Mo 3*d* XPS spectra; **b** Fe 2*p* XPS spectra; **c** P 2*p* XPS spectra; **d** Mo K-edge XANES spectra; **e** corresponding $k^2$-weighted FT of EXAFS spectra; **f** wavelet transforms for $k^2$-weighted EXAFS signals at Mo K-edge; **g** Fe K-edge XANES spectra; **h** corresponding $k^3$-weighted FT of EXAFS spectra; and **i** wavelet transforms for $k^3$-weighted EXAFS signals at Fe K-edge.

and no Fe–Fe peak is present at 2.47 or 2.85 Å in the Fourier-transform-EXAFS (Fig. 3h), suggesting that the Fe is largely isolated in Fe/SAs@Mo-based-HNSs, and is mostly coordinated with O (Fig. 3i and Supplementary Fig. 18)[34,35]. Specifically, one Fe atom is coordinated with 5.9 O atoms, and the mean Fe–O bond distance is 1.98 Å (Supplementary Fig. 18 and Supplementary Table 4). The XPS peak centred at 130.39 and 129.54 eV in the P 2*p* region (Fig. 3c) represents P bonded to Mo in MoP and $MoP_2$, while the P $2p_{3/2}$ peak at 134.18 eV represents the typical oxidation peak of P in residual phosphate[36]. The XPS peak at 454.7 eV in the O 1*s* spectrum corresponds to $O^{2-}$ (Supplementary Fig. 19), confirming the presence of $MoO_2$. The Fe content in the FeMoP-*T* catalysts substantially decreased as the annealing temperature was increased (Supplementary Table 5–7), which suggests slight evaporation of Fe species at high pyrolysis temperatures. Although the greater loss of Fe species in a higher pyrolysis temperature (e.g., in 550 °C), the residual Fe atoms were also in the form of monoatomic dispersion (Supplementary Fig. 20, Supplementary Table 8).

## Electrocatalytic alkaline HER performances

To highlight the electrocatalytic advantages of the synthetic heterostructured nanosheets, the performance of Fe/SAs@Mo-based-HNSs in the HER was investigated using a typical three-electrode system in 1 M KOH solution. The linear sweep voltammograms shown in Fig. 4a and

Supplementary Fig. 21a illustrate that the Fe/SAs@Mo-based-HNSs exhibited excellent performance in the HER: an overpotential of 38.5 mV vs reversible hydrogen electrode (RHE) at 10 mA cm$^{-2}$, which is much lower than those of other FeMoP-T samples and commercial 20 wt% Pt/C. The overpotential of Fe/SAs@Mo-based-HNSs slightly increase to 109.9 mV when the current density up to 200 mA cm$^{-2}$, significantly lower than that of commercial 20 wt% Pt/C (299.3 mV). The polarization curves of three independent Fe/SAs@Mo-based-HNSs prepared from different batches almost overlapped (Supplementary Fig. 22), indicating good repeatability of Fe/SAs@Mo-based-HNSs. The electrocatalytic HER mechanisms of the Fe/SAs@Mo-based-HNSs were examined by constructing Tafel plots (Fig. 4b, Supplementary Fig. 21b, and Supplementary Table 9), wherein it can be seen that they have a smaller slope (35.6 mV dec$^{-1}$) than those of bulk FeMoP-500 (89.3 mV dec$^{-1}$) and other FeMoP-T samples, close to that of 20% Pt/C (36.1 mV dec$^{-1}$). This indicates that a fast Heyrovsky step dominates the Volmer–Heyrovsky process during Fe/SAs@Mo-based-HNSs-electrocatalysed $H_2$ evolution, which is attributable to the strong electronic coupling within the single dispersed atoms and heterostructure.

Low charge-transfer resistance is a characteristic of an effective electrocatalyst, especially a C-free electrocatalyst[37,38]. Thus, electrochemical impedence spectroscopy was conducted at an overpotential of 200 mV to obtain charge-transfer data. As shown in Fig. 4c, the Fe/SAs@Mo-based-HNSs have a much smaller semicircular

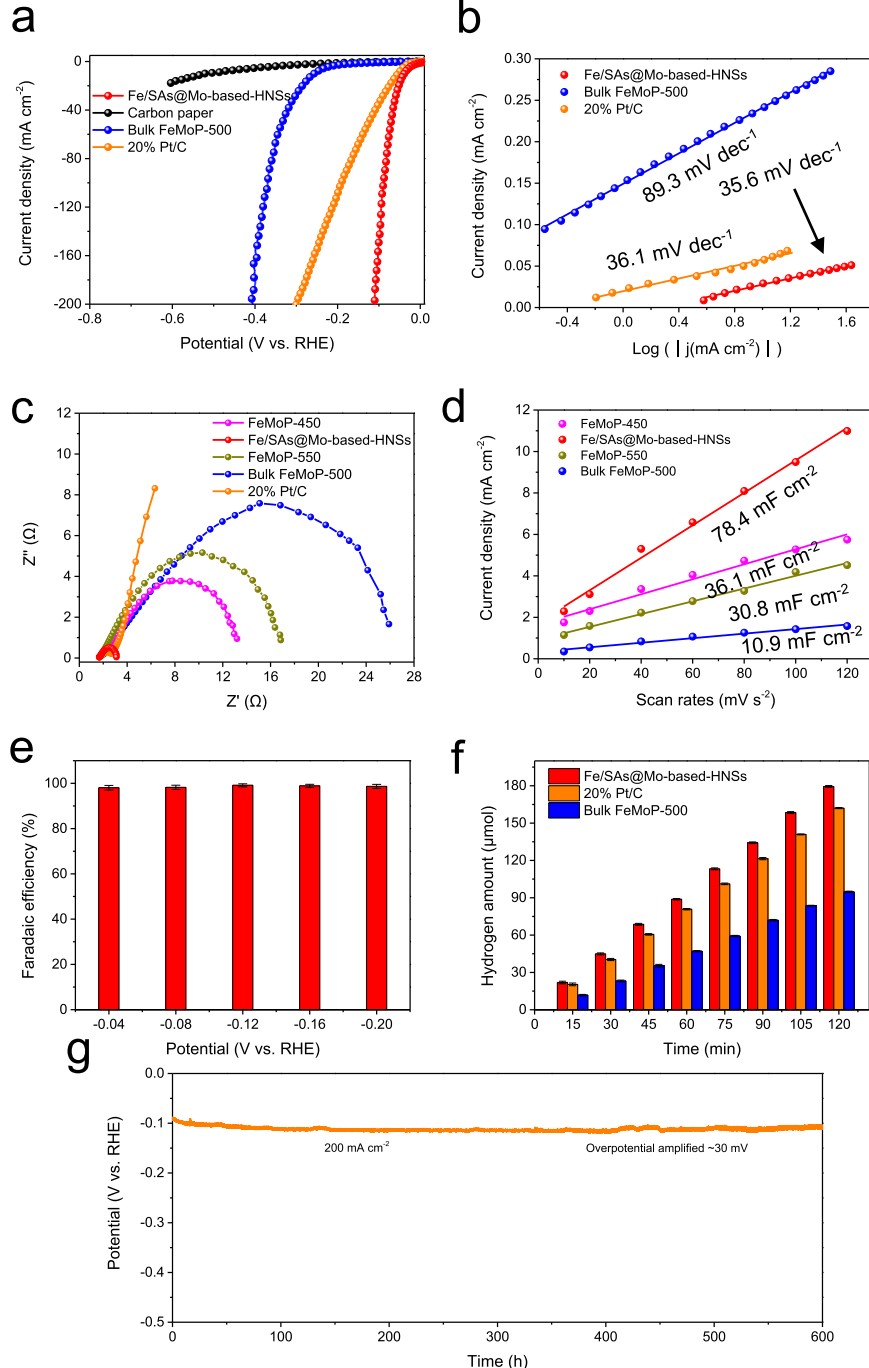

**Fig. 4 | Electrocatalytic performance of Fe/SAs@Mo-based-HNSs in the HER in 1.0 M KOH solution. a** Polarisation curves at a scan rate of 5 mV s$^{-1}$ with potential error (iR) correction; **b** corresponding Tafel plots; **c** Electrochemical impedance spectroscopy (EIS) of the FeMoP-T samples and 20% Pt/C at an overpotential of 200 mV; **d** calculated electrochemical double-layer capacitance for selected as-prepared materials; **e** Faradaic efficiency of Fe/SAs@Mo-based-HNSs at different applied potentials. The error bar reflects the three device results; **f** Hydrogen production at a specific constant current density of 10 mA/cm$^2$. The error bar reflects the three device results; **g** long-duration curves at a constant current density of 200 mA cm$^{-2}$.

diameter than the other phosphide samples, which is very similar to that of 20 wt% Pt/C; this suggests that the unique heterostructure with single-atom dispersed nature of Fe SAs@Mo-based HNSs, in addition to their 2D porous nanosheet morphology, contribute substantially to their optimised electronic structure. As variation in the surface area of electrocatalysts may affect their HER performance, the electrochemical surface areas of these electrocatalysts were estimated by measuring their electrochemical double-layer capacitances ($C_{dl}$) by cyclic voltammetry, which is typically considered representative of electrochemically active surface area (Supplementary Fig. 23). The $C_{dl}$

of Fe/SAs@Mo-based-HNSs was found to be 78.4 mF cm$^{-2}$ (Fig. 4d), which is much higher than that of the directly phosphatised bulk FeMoP-500 (10.9 mF cm$^{-2}$) and those of the FeMoP-450 and FeMoP-550 samples (36.1 and 30.8 mF cm$^{-2}$, respectively). Moreover, although the Fe/SAs@Mo-based-HNSs have a similar morphology, structure, and specific surface area to FeMoP-450 and FeMoP-550, the catalytic current density of Fe/SAs@Mo-based-HNSs at an overpotential of 100 mV is 14.2 times that of FeMoP-450, which is in turn only 3.6 times that of the bulk FeMoP-500. This suggests that the high HER electrocatalytic activity of Fe/SAs@Mo-based-HNSs is largely attributable to

the optimised electronic coupling in their abundant heterostructured active sites, and is supported by the very high electrolyte-accessible surface area of their 2D porous networks. Subsequently, gas chromatography was employed to detect the $H_2$ production and to determine the Faradaic efficiency. As can be seen in Fig. 4e, the Faradaic efficiency of Fe/SAs@Mo-based-HNSs is nearly 100% under a wide range of potentials. Controlled-current electrolyses at −10 mA cm$^{-2}$ were further conducted to compare the Faradaic efficiency with commercial 20 wt% Pt/C and bulk FeMoP-500, wherein it can be seen that the yield of $H_2$ of Fe/SAs@Mo-based-HNSs increases linearly over time and is significantly higher than that of commercial 20 wt% Pt/C and bulk FeMoP-500 (Fig. 4f and Supplementary Table 10). The Faradaic efficiency of Fe/SAs@Mo-based-HNSs after a 2-h period is 99.26%, which is higher than those of commercial 20 wt% Pt/C (89.46%) and bulk FeMoP-500 (52.74%).

The stability of the Fe/SAs@Mo-based-HNSs was also found to be excellent: they maintained a current density of 200 mA cm$^{-2}$ in the HER over a 600 h durability test, which represents ultrareliable performance. Specifically, as shown in Fig. 4g, the polarisation potential of Fe/SAs@Mo-based-HNSs exhibits little amplification of overpotential, that is only ~30 mV decreases at 200 mA cm$^{-2}$. The Supplementary Fig. 24 depicts the initial and post-500 h test polarisation curves of Fe/SAs@Mo-based-HNSs at 20 mA cm$^{-2}$, showing that they are extremely similar except in the area of the boosted onset potential. Moreover, the TEM (Supplementary Fig. 25), XRD (Supplementary Fig. 26), and XPS measurements (Supplementary Fig. 27) show that the Fe/SAs@Mo-based-HNSs retained porous multinary heterostructures with disordered edges after this durability test at 20 mA cm$^{-2}$, demonstrating that their heterostructure and valence states were well preserved and attesting to their excellent stability during the HER. The concentrations of elemental Fe, Mo, and P dissolved in solution after the durability test were measured by inductively coupled plasma–optical emission spectrometry (ICP-OES) (Supplementary Table 11). The concentration of all elements was close to the limit of reporting for the ICP-OES instrument, which is in accordance with the only slight change in the elemental composition of Fe/SAs@Mo-based-HNSs seen in the XPS analysis (Supplementary Table 5), and further illustrates the high stability of Fe/SAs@Mo-based-HNSs.

To further highlight the excellent electrocatalytic performance of our Fe/SAs@Mo-based-HNSs, we compared their overpotentials at a current density of 10 mA cm$^{-2}$ and the corresponding Tafel slope with those of various other electrocatalysts that have been reported for use in the HER in an alkaline medium (more details presented in Table S12). We found that the overpotential and the Tafel slope of the Fe/SAs@Mo-based-HNSs were superior to those of many non-precious-metal HER electrocatalysts and comparable to that of most of noble metal-based electrocatalysts. Thus, the ultrastability of our novel C-free porous 2D heterostructured Mo-based electrocatalysts may indicate they electrocatalyse the HER in alkaline media via a new and as-yet underdeveloped mechanism. If so, this could possibly be exploited for the design of high-performance HER electrocatalysts.

## Theoretical investigations

Previously reported heterostructured and single atom-dispersed HER electrocatalysts have been based on either active N-coordinated or C-coordinated metal sites, and no experimental or theoretical studies have been published describing O-coordinated single metal-atom dispersion heterostuctured HER electrocatalysts. We thus performed systematic DFT calculations to determine the molecular basis of the outstanding electrocatalytic activity of Fe/SAs@Mo-based-HNSs in the HER. It is well known that a key factor in the HER performance of electrocatalysts is the magnitude of the energy required for the adsorption of water molecules onto catalytic sites ($\Delta E_{H2O}$), especially in alkaline media[39]. Thus, $\Delta E_{H2O}$ values of various sites extracted from our experimental observations were calculated. This involved the

construction of three single-phase models (MoP, $MoP_2$, $MoO_2$), three heterostructured interfacial models ($MoP/MoP_2$, $MoP/MoO_2$, $MoP_2/MoO_2$), and two monoatomic-dispersed models (Fe@$MoO_2$−1, Fe@$MoO_2$−2), based on the microstructures we identified (Figs. 2 and 3). The results presented in Fig. 5a clearly show that all of the interfacial models have higher $\Delta E_{H2O}$ values than the single-phase models proving that the heterostructured interfaces effectively promote $H_2O$ adsorption. Moreover, the substantially different $H_2O$ adsorption abilities of monoatomic dispersed Fe atoms bonded with one (Fe@$MoO_2$−1) or two (Fe@$MoO_2$−2) O atoms in $MoO_2$ indicate that the local bonding environment of monoatomic dispersed Fe atoms greatly affects the $H_2O$ adsorption behaviour of electrocatalytic sites[40–42]. Furthermore, both the single-phase and heterostructured interfacial models show stronger $H_2O$ adsorption ability than Pt (111), demonstrating that MoP, $MoP_2$, and $MoO_2$ nanocrystals and the heterostructured interfaces are critical for efficient $H_2O$ adsorption. To further explain the principle of enhancing $H_2O$ molecule adsorption, the DFT simulations of atomic configurations and the corresponding electron density difference after $H_2O$ adsorption at Mo or Fe sites are shown in Fig. 5b. The $H_2O$ adsorption energy, adsorption site and the bonding distance between active site and O atom in $H_2O$ was summarized in Supplementary Table 13. These results show that the heterostructured interfaces and single Fe atom sites presents a significant depletion of electrons when adsorbing the $H_2O$ molecules, demonstrating the effective promotion of electrons transfer from active sites to $H_2O$ molecules[43]. In addition, the partial density of states (PDOS) of Mo/Fe active sites and O atoms in $H_2O$ molecules was calculated (Supplementary Fig. 28). The PDOS at s-, p-, d-orbitals of Mo/Fe and the s-, p-orbitals of O at heterostructured interface models are rather different compared to the single-phase models. This result explains the variation in the electron-transfer ability of different structures, and thus highlights the microstructures that are critical for efficient $H_2O$ adsorption.

As an important rate-determining factor for HER[44–46], the $H_2O$ dissociation performance on the surfaces of the above constructed models are further investigated and compared with Pt(111) (Supplementary Fig. 29)[46]. The $H_2O$ dissociation energies, either thermodynamically upward with low energy barrier or thermodynamically downward, indicate that $H_2O$ dissociation is easy to occur on the surface of catalysts. The results show that the $H_2O$ dissociation on the surface sites of $MoP_2$, $MoP/MoO_2$ and $MoP_2/MoO_2$ are thermodynamically upward, but much lower than that on the surface of Pt(111), while the others, especially $MoP/MoP_2$ and Fe@$MoO_2$−1, are thermodynamically downward, indicating the $H_2O$ dissociation is easy to occur on the surface of our catalyst in which $MoP/MoP_2$ and Fe@$MoO_2$−1 shows the greatest $H_2O$ dissociation capability. Combined with the excellent $H_2O$ adsorption energy of $MoP/MoP_2$ and Fe@$MoO_2$−1 (Fig. 5a), it can be concluded that the origin of efficient $H_2O$ adsorption and dissociation capability of our catalyst owes in part, to $MoP/MoP_2$ and Fe@$MoO_2$−1 promoting proton transfer to accelerate the Volmer step of HER.

To examine the second H* adsorption/$H_2$ desorption step (Heyrovsky step), the Gibbs free energies ($\Delta G_{H*}$) for all possible active sites were calculated (Fig. 5c), and the corresponding atomic configurations are shown in Fig. 5d and Supplementary Fig. 30. $\Delta G_{H*}$ is a key descriptor of HER performance under alkaline conditions, and HER activity increases as $\Delta G_{H*}$ approaches zero[47,48]. Figure 5c shows that the $\Delta G_{H*}$ values of H* adsorption onto all of the active sites in single-phase models are out of the range of −0.3–0.3 eV, indicating poor HER performance. In contrast, the $\Delta G_{H*}$ values for most of the active sites in heterostructured interfacial and monoatomic dispersed models are in the range of −0.19–0.22 eV, except for $MoP/MoP_2$. In particular, both the <Mo top> and <Mo-P bridge> sites in $MoP_2/MoO_2$ have $\Delta G_{H*}$ values very close to zero (0.04 eV and −0.05 eV) (Fig. 5d). Thus, the heterostructured interfaces and monoatomic dispersed sites underpin the excellent HER performance of the Fe SAs@Mo-based HNSs.

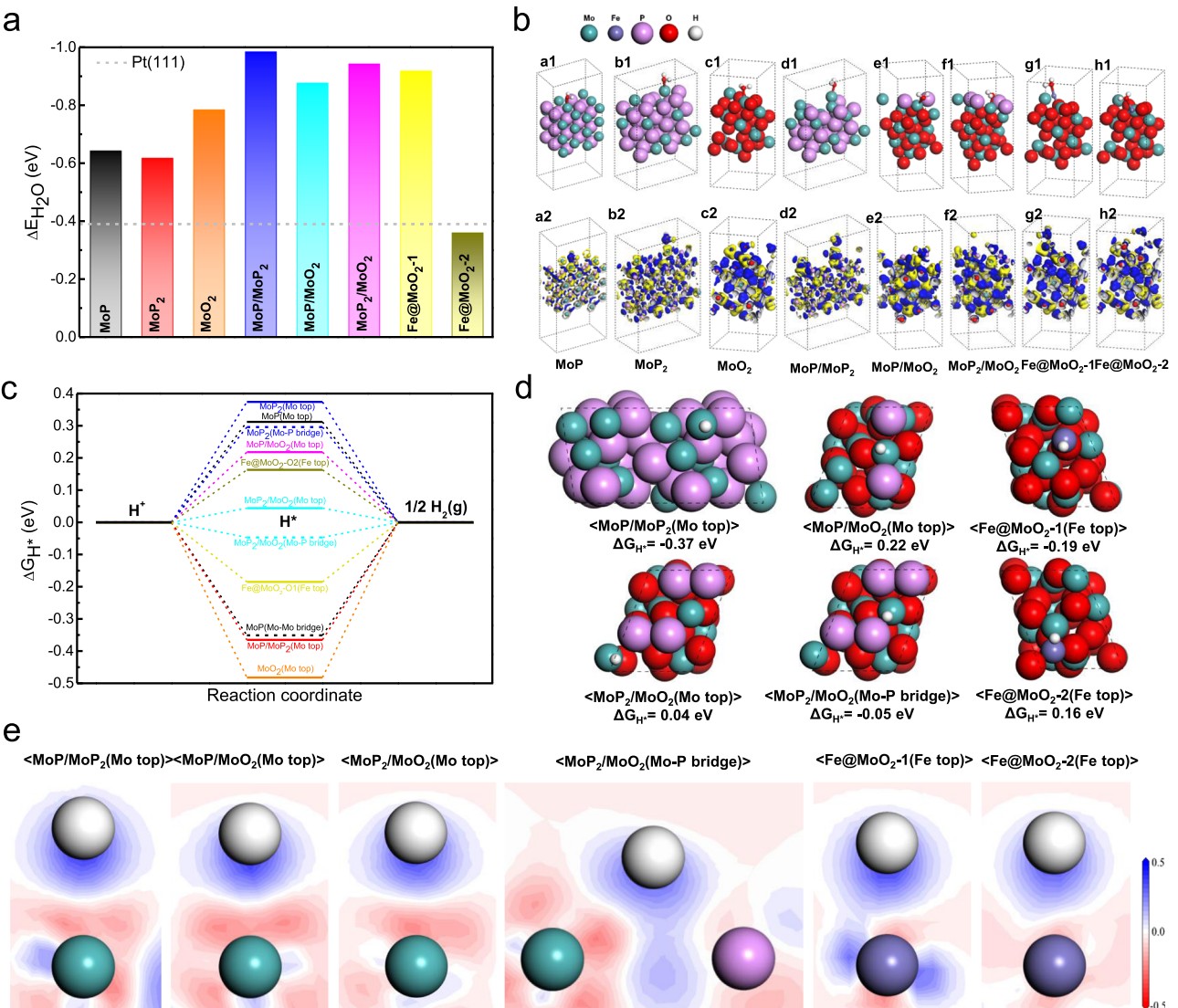

**Fig. 5 | DFT simulations of Fe/SAs@Mo-based-HNSs designed to catalyse HER in alkaline conditions. a** DFT-calculated $\Delta E_{H2O}$ on various exposed surfaces of MoP, MoP$_2$, MoO$_2$, MoP/MoP$_2$, MoP/MoO$_2$, MoP$_2$/MoO$_2$, Fe@MoO$_2$−1, and Fe@MoO$_2$−2, respectively. As a reference, the $\Delta E_{H2O}$ of the Pt(111) surface is marked by the grey dashed line. **b** DFT-optimised atomic configurations and corresponding electron density differences for MoP (a1 and a2), MoP$_2$ (b1 and b2), MoO$_2$ (c1 and c2), MoP/ MoP$_2$ (d1 and d2), MoP/MoO$_2$ (e1 and e2), MoP$_2$/MoO$_2$ (f1 and f2), Fe@MoO$_2$−1 (g1 and g2), and Fe@MoO$_2$−2 (h1 and h2), after H$_2$O adsorption at their surface sites. Yellow isosurfaces and blue isosurfaces represent the depletion and segregation of electrons, respectively. **c** $\Delta G_{H*}$ profiles of various catalytic sites at the surfaces of MoP, MoP$_2$, MoO$_2$, MoP/MoP$_2$, MoP/MoO$_2$, MoP$_2$/MoO$_2$, Fe@MoO$_2$−1, and Fe@MoO$_2$−2. **d** Representative atomic configurations after H* adsorption at the surface sites of MoP/MoP$_2$, MoP/MoO$_2$, MoP$_2$/MoO$_2$, Fe@MoO$_2$−1, and Fe@MoO$_2$−2, with corresponding $\Delta G_{H*}$. **e** DFT-calculated 2D electron density differences after adsorption of H* on active sites in heterostructured interface models and monoatomic dispersed models. Red backgrounds and blue backgrounds represent the depletion and accumulation of electrons (e/Å$^3$), respectively.

To clarify the fundamental physical mechanism of H* adsorption, the magnitude of electron transfer between H* and active sites was analysed by investigating the 2D electron density differences resulting from H* adsorption onto the surface sites of single-phase (Supplementary Fig. 31), heterostructured interface, and monoatomic dispersed models (Fig. 5e). In single-phase models, the electron transfer between H* and active sites that is either too strong or too weak, which results in these models having poor H* adsorption/desorption abilities. While in the heterostructured interface models, the electron transfer from the Mo top at the MoP$_2$/MoO$_2$ interface to H* is between that of MoP/MoP$_2$ ($\Delta G_{H*} = -0.37$ eV) and MoP/MoO$_2$ ($\Delta G_{H*} = 0.22$ eV), and results in an ideal $\Delta G_{H*}$ value (0.04 eV). A comparison of <MoP$_2$ (Mo-P bridge)> ($\Delta G_{H*} = 0.30$ eV) and <MoP$_2$/MoO$_2$ (MoP bridge)> ($\Delta G_{H*} = -0.05$ eV) shows that the electron transfer from Mo to H* is greatly promoted at the MoP$_2$/MoO$_2$ interface. In the Fe monoatomic

dispersed models, we found that the electron transfer from Fe to H* is greater at the Fe site in Fe@MoO$_2$−1 than that in Fe@MoO$_2$−2. This is consistent with the difference in their $\Delta G_{H*}$ values (−0.19 eV for Fe@MoO$_2$−1 and 0.16 eV for Fe@MoO$_2$−2). These favourable $\Delta G_{H*}$ values indicate that monoatomic dispersed Fe atoms bonded with either one or two O atoms in MoO$_2$ possess appropriate H* adsorption/ desorption ability for effective HER electrocatalysis.

To further explain the excellent HER performance of these sites, the d-orbital PDOS was also calculated (Supplementary Fig. 32). The results clearly show that the centres of d-orbital PDOSs of Mo sites in MoP(−1.23 eV) and MoP$_2$ (−1.40 eV) are either too far away from, or in MoO$_2$ (−0.75 eV) are too close to the Fermi level, resulting in too weak or too strong interactions between catalytic sites and H*. In contrast, the centres of d-orbital PDOSs of Mo sites in MoP/MoO$_2$ and MoP$_2$/ MoO$_2$ interfaces have appropriate values (−1.13 eV and −0.99 eV,

respectively), which account for their better $\Delta G_{H^*}$ values. Moreover, the change in the centres of d-bands is consistent with the variation in $\Delta G_{H^*}$ values (Supplementary Table 14) indicate that the appropriate d-band centre for Mo top sites is -1.0 eV. These results revealed that the Heyrovsky reaction occurs most efficiently at $MoP_2/MoO_2$ interfaces and monoatomic dispersed Fe locations. This is perfectly consistent with the experimental results that the best HER performance of Fe/SAs@Mo-based-HNSs was achieved at an appropriately 'medium' temperature (500 °C), owing to its highest ratio of $MoP_2/MoO_2$ interfaces and monoatomic dispersed Fe locations.

Based on the DFT and experimental results and analysis, the superior HER performance of the Fe/SAs@Mo-based-HNSs is attributable to the following reasons. (1) The heterostructured interfaces of Fe/SAs@Mo-based-HNSs lead to optimised electronic structures and H* adsorption energies, increasing their intrinsic electrocatalytic activity. (2) The monoatomic dispersed Fe locations contribute to efficient $H_2O$ adsorption and dissociation capability, promoting proton transfer to accelerate the HER performance. (3) The unique 2D and hierarchical porous architecture of Fe/SAs@Mo-based HNSs leads to effective mass transport within their structures and high exposure of active sites to electrolyte, thereby enhancing their electrocatalytic activity. Fe/SAs@Mo-based-HNSs is the best-performing single-atom dispersed heterostructured HER electrocatalyst.

## Discussion

In summary, we have devised a highly efficient HER electrocatalyst composed of porous Fe/SAs@Mo-based-HNSs, which is formed via a novel low-temperature phosphorisation of environmentally benign and simple self-assembled inorganic–inorganic coordinated FePMoG nanosheets. The porosity and the heterogeneous interfaces of the heterostructured nanosheets derived from the FePMoG nanosheets were easily regulated during the phosphorisation process by adjusting the temperature and time. Consequently, our Fe/SAs@Mo-based-HNSs exhibited an overpotential of 38.5 mV at a current density of 10 mA cm$^{-2}$, a Tafel slope of 35.6 mV dec$^{-1}$, and long-term durability with negligible increase in polarisation potential over 600 h of continuous operation at 200 mA cm$^{-2}$, which is one of the best HER performances reported to date for nanostructured Mo-based electrocatalysts. This outstanding performance is attributable to the Fe/SAs@Mo-based-HNSs' optimised electronic structure, enriched interface and boundary phases, large active surface areas and porosities, and the synergetic effect of their single dispersed atoms and heterostructures. Experiments and DFT calculations reveal that the heterostructured interface and the single dispersed Fe atoms of Fe/SAs@Mo-based-HNSs facilitate $H_2O$ adsorption and appropriate H* adsorption/desorption during the HER, and their porous 2D network means that their active sites are highly accessible.

Thus, this work describes a simple, new approach for fabricating 2D nanosheet precursors of nanostructured C-free porous heterostructured nanosheets, and for optimising their electronic structures via a multi-phase strategy to increase their electrocatalytic performance in the HER. This high-performance C-free electrocatalyst represents an effective alternative to corrosion-prone C-containing catalysts for use in proton-exchange membrane fuel cells and other state-of-the-art energy technologies.

## Methods

### Preparation of wrinkle-like FePMoG nanosheets

In a typical procedure, 12 ml of aqueous $Fe(NO_3)_3 \cdot 9H_2O$ (0.5 M) solution was added to 30 ml phosphomolybdic acid (PMo, 15 mg ml$^{-1}$), and the solution was kept still for a designated time. As the reaction proceeded, the transparent yellow solution gradually became turbid due to the formation of nanoparticles or nanosheet self-assembled by $Fe^{3+}$ and PMo. After reacting for 20 h, the suspension was centrifuged and washed with distilled water three times. The centrifugation procedure (Centrifugation speeds: 447 g) is used to remove the excessive water toward the formation of FePMoG mineral hydrogel. The morphology of $Fe^{3+}$-phosphomolybdic acid assemblies can be easily controlled by changing the reactant concentration.

### Preparation of Fe/SAs@Mo-based-HNSs

The as-obtained wrinkle-like FePMoG mineral hydrogel was redispersed in 30 ml ethanol and 30 g NaCl was added into the dispersion under magnetic stirring to avoid the aggregation of the nanosheets. The ethanol was evaporated under 60 °C with continuous magnetic stirring and obtained the FePMoG and NaCl mixture (denoted as FePMoG-NaCl). The dried FePMoG-NaCl was annealed at 500 °C in Ar gas flow for 2.5 h with a heating rate of 3.5 °C min$^{-1}$ with the sodium hypophosphite in the upstream as phosphorus source and to produce the Fe/SAs@Mo-based-HNSs. For comparison, the control samples were prepared at different annealing temperature.

### Characterizations

The samples were characterized with scanning electron microscopy (SEM, Philips XL-30 FESEM) and high-resolution transmission electron microscopy (TEM, JEOL TEM 2100F FEG operated with an accelerating voltage of 200 kV). Atomic-resolution scanning transmission electron microscopy (STEM) images and energy-dispersive spectrometry (EDS) maps were acquired on a Titan FEI Themis G60-300 S/TEM (fitted with a high-brightness field emission gun (X-FEG), probe Cs corrector and super-X EDS with four windowless silicon drift detectors). The X-ray diffraction (XRD) patterns were collected using an X-ray diffractometer (Rigaku SmartLab) with Cu Kα ($\lambda = 1.5418$ Å). Fourier transform infrared spectroscopy (FT-IR) spectra were documented with KBr pellets from a Bruker Model R 200-L spectrophotometer. Brunauer-Emmett-Teller (BET) experiments were tested on a Quantachrome NOVA 1200e Gas Adsorption Analyzer at 77 K. X-ray photoelectron microscopy (XPS) was performed on a ESCALAB 250 photoelectron spectrometer (ThermoFisher Scientific) with Al Ka (1486.6 eV) as the X-ray source set at 150 W and a pass energy of 30 eV for high-resolution scan, The base pressure was $3 \times 10^{-9}$ mbar, and the binding energies were referenced to C1s line at 284.8 eV from adventitious carbon. The Fe/Mo K-edge X-ray absorption near edge structure (XANES) and extended X-ray absorption fine structure (EXAFS) experiments for the Fe/SAs@Mo-based-HNSs were carried out at BL07A1 beamline of National Synchrotron Radiation Reaearch Center (NSRRC). The data were collected in fluorescence mode using a Lytle detector while the corresponding reference sample were collected in transmission mode. The incident beam was monochromatized by using a Si (111) fixed-exit, double-crystal monochromator, and a harmonic rejection mirror was applied to cut off the high-order harmonics. The obtained XAFS data was processed in Athena (version 0.9.26) for background, pre-edge line and post-edge line calibrations. Then Fourier transformed fitting was carried out in Artemis (version 0.9.26). The k$^3$ weighting, k-range of 3–14 Å$^{-1}$ and R range of 1–~3 Å were used for the fitting of Mo/Fe foil; k-range of 3–12 Å$^{-1}$ and R range of 1–~3.5 Å were used for the fitting of Mo samples, k-range of 3–10 Å$^{-1}$ and R range of 1- -2 Å were used for the fitting of Fe samples. The four parameters, coordination number, bond length, Debye-Waller factor, and E$_0$ shift (CN, R, $\Delta E_0$) were fitted without anyone was fixed, the $\sigma^2$ was set. For Wavelet Transform analysis, the χ(k) exported from Athena was imported into the Hama Fortran code. The parameters were listed as follow: R range, 1–4 Å; k range, 0–13 Å$^{-1}$ for Mo and 0–15 Å$^{-1}$ for Fe; k weight is 2 and 3 for Mo and Fe, respectively; and Morlet function with κ = 3, σ = 1 was used as the mother wavelet to provide the overall distribution.

### Electrochemical measurements

Prior to use, the carbon paper was cut into cut into small pieces, then the carbon paper chips were washed ultrasonically in ethanol and

dried in oven. First, 4 mg catalyst was ultrasonically dispersed in 400 μL of 0.5 wt% Nafion solution (ethanol: water = 3: 1) to form homogeneous slurry. The slurry was then coated onto a carbon paper with a catalyst loading of 670 μg cm$^{-2}$. The catalyst-coated carbon paper electrode, saturated calomel electrode (SCE), and graphite rod were used as the working, reference, and counter electrodes, respectively. Electrochemical measurements were conducted at room temperature on a CHI760e electrochemical station with a three-electrode cell system. Linear sweep voltammetry was recorded in H$_2$-saturated 1 M KOH at a scan rate of 5 mV s$^{-1}$ to obtain the polarization curves. The long-term stability tests were performed by continuously applying a current density of 200 mA cm$^{-2}$ to the working electrode and used a Hg/HgO to alter the SCE as reference electrode (The H$_2$ wasn't supplied in the stability test as the H$_2$ constantly being produced in the working electrode therefore the electrolyte in the working electrode side can be considered as H$_2$ saturated). All the data presented were corrected for iR losses and background current, and the potentials were later converted to the reversible hydrogen electrode (RHE) scale (for SCE reference in 1 M KOH, V$_{RHE}$ = V$_{SCE}$ + V$_{SCE}^{\theta}$ + 0.059 pH). Cyclic voltammograms (CV) were obtained with different scan rates from 4 to 40 mV s$^{-1}$ in the potential range of 0-0.1 V (vs. RHE).

The Faradaic efficiency (FE) of catalysts is defined as the ratio of the amount of experimentally determined H$_2$ to that of the theoretically expected H$_2$ from the reaction, which was measured at different potentials (−0.04, −0.08, −0.12, −0.16 V, and −0.20 vs. RHE) and at a specific constant current density of 10 mA/cm$^2$ by gas chromatography (GC-2014, Shimadzu). The experiments were conducted in a custom-made two compartment cell (single cell: 50 mL) separated by a Nafion 117 membrane, each compartment of the cell was filled with 35 mL 1.0 m KOH. The H$_2$ gas was purged out from the cell by using 100 μL syringe and injected into gas chromatography. For the specific applied potentials, the gas was sampled after one hour potential supply; for the specific constant current density of 10 mA/cm$^2$, the gas was took a sample every 15 min. FE was calculated according to following relationship

$$FE = \frac{2F \times n_{H_2}}{Q}$$

Where, n$_{H2}$ is the amount of hydrogen (mol), F is the Faraday's constant (96485 C mol$^{-1}$), and $Q$ is the total amount of charge passed through the cell (C).

## DFT calculations

Density functional theory (DFT) calculations, as one of the most versatile tools to study electrocatalytic mechanism, are conducted to investigate hydrogen evolution reaction (HER) behaviors of Fe/SAs@Mo-based-HNSs using the Cambridge Sequential Total Energy Package (CASTEP) module in Materials Studio software[49]. The generalized gradient approximation (GGA) method and Perdew-Burke-Ernzerh (PBE) function are employed to describe the exchange and corrections of interaction between atoms[50]. The ultrasoft pseudopotential method is utilized to describe the interactions between valence electrons and ionic cores[51]. A plane-wave basis set is assigned with a cutoff energy of 400 eV. The Brillouin zone is sampled by a Monkhorst-Pack grid[52]. The tolerances of energy, force and displacement for structure optimization are 10$^{-6}$ eV/atom, 0.02 eV/Å and 0.001 Å respectively. The self-consistence field (SCF) is set as 5×10$^{-6}$ eV/atom. The effect of van der Waals interaction is taken into account by the semi-empirical DFT-D force-field approach[53].

The Gibbs free energies for hydrogen adsorption (ΔG$_{H*}$) are calculated from the Eq. 1:

$$\triangle G_{H^*} = \triangle E_{H^*} + \triangle ZPE - T\triangle S \tag{1}$$

Where the $\triangle E_{H^*}$, $\triangle ZPE$, $T$ and $\triangle S$ represent the binding energy, zero-point energy change, temperature and entropy change of H adsorption system, respectively.

The vibration entropy is H at the adsorbed states is negligible. Thus, $\triangle S$ can be obtained from the following Eq. 2:

$$\triangle S = S_{H^*} - \frac{1}{2}S_{H_2} \approx -\frac{1}{2}S_{H_2} \tag{2}$$

Where $S_{H_2}$ is the entropy of H$_2$ in the gas phase at the standard conditions.

Besides, $\triangle ZPE$ can be calculated from the Eq. 3:

$$\triangle ZPE = ZPE_{H^*} - \frac{1}{2}ZPE_{H_2} \tag{3}$$

Thus, the free energy of the adsorbed state can be calculated using the simplified Eq. 4[48]:

$$\triangle G_{H^*} = \triangle E_{H^*} + 0.24\,eV \tag{4}$$

The water adsorption energies (ΔE$_{H2O}$) at the surface of catalysts are calculated by the Eq. 5:

$$E_{H_2O} = E_{surf+H_2O} - E_{surf} - E_{H_2O} \tag{5}$$

Where the $E_{surf}$ and $E_{surf+H_2O}$ are the total energies of the surface before and after water adsorption, and $E_{H_2O}$ is the energy of a free water molecule.

Therefore, the HER performance of catalysts can be evaluated by DFT calculations of ΔE$_{H2O}$ and ΔG$_{H*}$, which are the key descriptors for the Volmer-Heyrovsky reaction efficiency.

According to the experimental characterization information, a series of representative single-phase, heterostructured interface, and monoatomic dispersed models are built for DFT study of their HER performance with corresponding atomic mechanism investigation. Specifically, (1) MoP, MoP$_2$ and MoO$_2$ single-phase models; (2) MoP/MoP$_2$, MoP/MoO$_2$ and MoP$_2$/MoO$_2$ heterostructured interface models and (3) two kinds of monoatomic dispersed Fe onto MoO$_2$ surface models are constructed, bonding with one or two O atoms in MoO$_2$. The initial atomic configurations of single-phase MoP, MoP$_2$ and MoO$_2$ are from crystallography open database. In order to build the MoP/MoP$_2$, MoP/MoO$_2$ and MoP$_2$/MoO$_2$ interfaces, the heterogeneous plane structure consisting of four layers of MoP$_2$ or MoO$_2$ with (2×2) supercells in the bottom, with the addition of MoP or MoP$_2$ clusters onto the top of them[54]. Since the DFT calculation results are very sensitive to the local coordination structure of the metallic centers[40-42], a series of initial configurations with Fe atoms coordinating different number of O atoms are constructed to demonstrate the representative stable models with monoatomic dispersed Fe onto MoO$_2$. Two different stable monoatomic dispersed Fe onto MoO$_2$ surface models (bonding with one or two O atoms, named Fe@MoO$_2$−1 and Fe@MoO$_2$−2 respectively) are obtained via a series of structure optimization test. The constructed initial configurations with Fe atoms coordinating more O atoms are unstable. It should be noted that the monoatomic Fe sites only coordinate with 1 or 2 O atoms in our DFT models which is much smaller than the result obtained from EXAFS. However, it should be noted that the data obtained from EXAFS is the information of average coordination number, which mainly be determined by the coordination number of Fe atoms in MoO$_2$ interior. Since the Fe active sites contributing to HER are the monoatomic Fe atoms at the surface of the material, their coordination number should be smaller than that in the material interior, existing in the form of unsaturated coordination. Therefore, the DFT models only containing 1 or 2 O coordination atoms are taken as the representative

monoatomic Fe models. A vaccum gap of 15 Å is employed to create free surface for the simulation of hydrogen evolution reaction (HER) behaviors for all the models. The $k$ points are set as $2 \times 4 \times 1$ for MoP, $MoP_2$, $MoP/MoP_2$, and $5 \times 4 \times 1$ for $MoO_2$, $MoP/MoO_2$, $MoP_2/MoO_2$, $Fe@MoO_2-1$, $Fe@MoO_2-2$ to sample the Brillouin zone.

## Data availability

Source data are provided with this paper. The data used to generate the Figs. 4a, g shown in this study, and other two independent LSV test data for Fe/SAs@Mo-based-HNSs have been deposited. Further data supporting the findings of this study are available in the Supplementary Information. All other relevant source data are available from the corresponding author upon request. Source data are provided with this paper.

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

## Acknowledgements
This work is supported by Shenzhen-Hong Kong Science and Technol-ogy Innovation Cooperation Zone Shenzhen Park Project (HZQB-KCZYB-2020030) to J.L.; the National Key R&D Program of China (Project No. 2017YFA0204403) to J.L.; Hong Kong Innovation and Technology Commission via the Hong Kong Branch of National Precious Metals Material Engineering Research Center; the National Natural Science Foundation of China (Project No. 12002108) to L.S.; the Guangdong Basic and Applied Basic Research Foundation (Project No. 2020A1515110236, 2022A1515011402) to L.S.; and the Science, Tech-nology, and Innovation Commission of Shenzhen Municipality (Project No. GXWD20201230155427003-20200824105236001, ZDSYS20210616110000001) to L.S.

## Author contributions
F.L., Y.Y.L., and J.L. designed the research. S.Z., Z.J., F.M., and Z.M. prepared and characterised the samples. L.C. and J.P. conducted SEM and XPS. F.L., Y.B., and Y.B. conducted TEM. L.S. performed the theoretical calculations. F.L., S.Z., L.S., Z.J., Y.Y.L., and J.L. wrote the manuscript. All authors discussed the results and reviewed the manuscript.

## Competing interests
The authors declare no competing interests.
