## [Peer review file · Nature Communications]

REVIEWER COMMENTS

Reviewer #1 (Remarks to the Author):

In this paper, the authors synthesized porous Fe/SAs@Mo-based-HNSs electrocatalyst for HER by using self-assembly and phosphorisation process. Atomically dispersed Fe and heterostructures lead to achieving low overpotential and long term durability. The authors should address the following concerns before the recommendation of publication.

1. The FeP can be obtained under 500°C phosphorization temperatures. How the FeP nanoparticles transfer into atomically dispersed Fe that coordinated with O at 500°C?
2. In the line 257, the authors claim that “The Mo K edge XANES spectrum shows that the near edge absorption energy of Fe/SAs@Mo based HNSs is intermediate between that of Mo foil and MoO₂, demonstrating that the average oxidation state of Mo is between Mo³⁺ and Mo⁴⁺”. However, the average oxidation state of Mo between 0 and 4+ can be concluded. Because there is no reference with Mo³⁺ showed.
3. The XPS results show that Mo⁴⁺ and Mo⁶⁺ are the majority. Why the average oxidation state of Mo in XAS is lower than 4+?
4. In the line 293, “they have a smaller slope (35.1 mV dec⁻¹) than those of bulk FeMoP 500 and 20% Pt/C.” However, the Tafel slope of 20% Pt/C is 33.6 mV dec⁻¹ as showed in Figure 4b.
5. Some mistakes should be corrected. In the line 251, “synchrotron radiation based soft X ray absorption near edge structure (XANES)”. It should be hard XANES. In the line 283, “(h) corresponding k₂-weighted FT of EXAFS spectra;” this is different from “FTk₃χ(k)” in the Y axis of Figure 3h. Are they k₂- or k₃-weighted FT of EXAFS?

Reviewer #2 (Remarks to the Author):

Review comments:

This manuscript by Lyu et al. reported the fabrication of the single iron atom dispersed Mo-based nanosheet heterostructure developed from a mineral hydrogel. The heterostructured catalysts were applied as efficient hydrogen evolution reaction (HER) electrocatalysts, showing high activity (a small overpotential of 38 mV at 10 mA cm⁻²) and excellent durability (negligible performance decay after 500 h operation). Furthermore, theoretical calculations demonstrated the role of O-coordinated single Fe atoms in the heterostructure. Overall, I think this manuscript is well organized and written. However,

several concerns and questions need to be properly addressed before possible consideration for publication in Nature Communications.

1. The authors claimed this synthesis approach shows “universality”, but I cannot find any evidence in this work. It would be interesting to see how this method can be applied to other single metal dispersed structures.

2. If I understand correctly, the authors concluded MoP and MoP₂ as the most active sites for the HER, and the single dispersed Fe atoms enhance the activity of Mo-based catalysts by introducing additional structural vacancies. Here, the single-atom Fe seems to serve as spectators or absorbing *H₂O to trigger the reaction. However, first, if the binding energy of *H₂O on Fe@MoO₂-2 is very small, such weak adsorption cannot compete with other sites showing very strong binding capabilities; second, if Fe is not the active center, why does one need a sufficient amount of Fe to deliver a decent HER activity? Overall, the experiments and calculations do not match well in this work, or the current descriptions have not been well delivered. Possible synergistic effects may need to be further discovered here.

3. Followed by comment 2, the DFT calculations are weak and are hence not reliable to support the experimental conclusions. For example, the electrocatalysts are operated in an alkaline solution, but the DFT calculations are considered on the acidic Volmer step. The calculations can be complicated if one further considers the pH value effect, the solvent effect, and the reaction kinetics of key elementary steps. However, these aspects were not reflected in the current DFT works. The conclusions may be easily altered by different theoretical results.

4. Would other thermochemical treatment temperatures yield single metal dispersed structures?

5. How could you know there are no Fe-P bonds but only Fe-O? The coordination environment of the Fe atom is 5.9 O atoms from EXAFS, why do the DFT models only contain 1 or 2 O atoms? This is important because DFT calculation results are very sensitive to the local coordination structure of the metallic centers (see some related references: Small 2022, 18, 2105680; Adv. Mater. 2021, 33, 2103004; Mater. Today Energy 2021, 20, 100653).

6. In figure 3d, from the statement “The Mo K-edge XANES spectrum shows that the near-edge absorption energy of Fe/SAs@Mo-based-HNSs is intermediate between that of Mo foil and MoO₂, demonstrating that the average oxidation state of Mo is between Mo³⁺ and Mo⁴⁺”, one would agree the valence state of Mo locates in between Mo and Mo⁴⁺, but why in between Mo³⁺ and Mo⁴⁺? The more accurate determinations of valence states of elements are required based on the XANES results.

7. For the evaporation of Fe species at high pyrolysis temperatures, can you show more examples?

8. How would the effect of hydrogen spillover be here?

9. Some typos were found in the manuscript, not limited to:

“single atom subtrate precursors” should be “single atom substrate precursors”?

“the common substragte precursors” should be “the common substrate precursors”?

Reviewer #3 (Remarks to the Author):

Comments to the Author

In this article, the authors have devised a highly efficient HER electrocatalyst composed of porous Fe/SAs@Mo-based-HNSs, which is formed via a novel low-temperature phosphorisation of environmentally benign and simple self-assembled inorganic–inorganic coordinated FePMoG nanosheets. The author attribute to the good performance of HER to Fe/SAs@Mo-based-HNSs' optimised electronic structure, enriched interface and boundary phases, large active surface areas and porosities, and the synergetic effect of their single dispersed atoms and heterostructures. However, there are some problems in the article, and the experimental data cannot fully support this result. After making the following major revisions, this manuscript can be published in the *nature communication*.

In Figure 1a, the author makes a quantitative comparison of several substrate precursors. But there is no evidence to support this comparison. And this figure shows that the mineral hydrogel seems to be a too perfect substrate precursor, because every index is optimal, I think this figure is kind of misleading.

The synthetic diagram of the material (figure 1b) is too simple, and many important information are not shown. For example, this diagram does not reflect how Fe single atoms are formed, and even the reaction conditions and precursors are not shown.

Why is Fe SAs only formed by pyrolysis at 500°, and whether Fe also exists in the form of single atoms at other temperatures? This needs to be confirmed. If Fe SAs can also be formed at other temperatures, the article mentions that these single-atom dispersed heterostructured nanosheets was first developed from a mineral hydrogel, but the performance of bulk FePMo-500 is even better than that of FePMo-450, which may prove that Fe single atoms are not very important for the performance of HER. In addition, from figure s21, the Fe content seems to become very little in the samples after a long time of testing, does this also indicate that Fe SAs is not an active site for HER.

The article said “high HER electrocatalytic activity of Fe/SAs@Mo-based-HNSs is largely attributable to the optimised electronic coupling in their abundant heterostructured active sites, and is supported by the very high electrolyte-accessible surface area of their 2D porous networks”, but from figure s11 and s12, There are also many heterostructured active sites in FeMoP-450 and FeMoP-550, so it is not convincing.

There are too many abbreviations in the article. For example , FePMoGs , FePMoG, ,FePMo and FePMo-T can easily cause confusion in reading.

Responses to the reviewers' comments

Ms. Ref. No.: NCOMMS-22-13743

Title: "2D mineral hydrogel-derived single atoms-anchored heterostructures for ultrastable hydrogen evolution "

First of all, we thank the reviewers for their careful reading of our manuscript and their constructive suggestions. Below, we list the reviewers' comments in **blue** text and our responses to each in **black** text. We have adopted every reviewer suggestion and we are now confident that our manuscript is suitable for publication in *Nature Communications*.

We have denoted our updates in the revised manuscript using **red** text for easy identification. Also, according to the formatting instructions, the title of the manuscript has been revised to "2D mineral hydrogel-derived single atoms-anchored heterostructures for ultrastable hydrogen evolution".

Reviewer #1

General Comments: In this paper, the authors synthesized porous Fe/SAs@Mo-based-HNSs electrocatalyst for HER by using self-assembly and phosphorisation process. Atomically dispersed Fe and heterostructures lead to achieving low overpotential and long-term durability. The authors should address the following concerns before the recommendation of publication.

Comments 1: The FeP can be obtained under 500°C phosphorization temperatures. How the FeP nanoparticles transfer into atomically dispersed Fe that coordinated with O at 500°C?

Response: We thank the reviewer for the constructive comments. In our opinion, Fe is coordinated with O of phosphomolybdic acid and is atomically dispersed in FePMoG, and the Fe atoms were gradually phosphorized during the phosphorisation process. When the temperature is above 500 °C, the transformed FeP would escape from the body material easily while the unphosphated Fe is still coordinated with O and therefore maintains the monatomic state. From Table R1, R2 and R3 (also shown in Supplementary Table 5-7 in revised Supplementary information), we can see that the Fe percentage of Fe/SAs@Mo-based-HNSs decreases compared with FePMoG. When the temperature is $\leq 450^\circ\text{C}$, the FeP phase can be detected by XRD, and the Fe percentage of FeMoP-450 is also lower than that of FePMoG, indicating the release of the formed FeP is limited under a relative low temperature and thus aggregated into the FeP nanoparticles. The ratio of $r_{\text{Fe}/\text{Mo}}$ is dramatically reduced

to a small value when the temperature raised to 550°C, indicating the easy release of FeP under this condition. The newly added Fe K-edge XANES results of FeMoP-550 also demonstrated that the Fe is monatomic dispersion (Fig. R1 and Table R4, also shown in Supplementary Fig. 19 and Supplementary Table 8 in revised Supplementary information). From the above analysis, we knew that the 500°C is a critical temperature in our reported system. Thus, we surmise that the Fe and Mo atoms are partially phosphorized and the formed FeP is released at 500°C, in the meanwhile, the maintained Fe atoms still keep coordinated with O and therefore obtained the atomically dispersed Fe.

Table R1. Elemental compositions for FePMoG, FeMoP-450, Fe/SAs@Mo-based-HNSs and FeMoP-550 determined by XPS.

Samples	Fe (at%)	Mo (at%)	P (at%)	O (at%)	$r_{\text{Fe/Mo}}$
FePMoG	7.8	16.9	0.4	74.9	0.461
FeMoP-450	4.5	18.8	8.1	68.6	0.239
Fe/SAs@Mo-based HNSs	4.6	20.7	10.4	64.3	0.222
FeMoP-550	1.0	17.3	19.8	61.9	0.058

$r_{\text{Fe/Mo}}$ is the atomic ratio of Fe to Mo.

Table R2. Elemental compositions for FePMoG, FeMoP-450, Fe/SAs@Mo-based-HNSs and FeMoP-550 determined by TEM EDS.

Samples	Fe (at%)	Mo (at%)	P (at%)	O (at%)	$r_{\text{Fe/Mo}}$
FePMoG	8.28	15.56	0.51	75.65	0.532
FeMoP-450	5.3	19.7	8.9	66.1	0.269
Fe/SAs@Mo-based HNSs	5.1	21.4	11.3	62.2	0.238
FeMoP-550	1.4	18.2	21.9	58.5	0.077

$r_{\text{Fe/Mo}}$ is the atomic ratio of Fe to Mo.

Table R3. Elemental compositions for FePMoG, FeMoP-450, Fe/SAs@Mo-based-HNSs and FeMoP-550 determined by ICP-OES.

Samples	Fe (mg/kg)	Mo (mg/kg)	P (mg/kg)	O (mg/kg)	$r_{\text{Fe/Mo}}$
FePMoG	146725.83	481927.19	/	/	0.523
FeMoP-450	70586.59	431511.55	/	/	0.281
Fe/SAs@Mo-based HNSs	62367.61	420141.41	/	/	0.255

FeMoP-550	22122.89	447094.81	/	/	0.085
-----------	----------	-----------	---	---	-------

$r_{\text{Fe/Mo}}$ is the atomic ratio of Fe to Mo.

Fig. R1 Spectroscopy of FeMoP-550 at Fe K-edge: (a) Fe K-edge XANES spectra; (b) corresponding k^3 -weighted FT of EXAFS spectra; (c) the corresponding k^3 -weighted FT-EXAFS spectra and fitting line in the R spacing; and (d) wavelet transforms for k^3 -weighted EXAFS signals.

Table R4. EXAFS fitting parameters at the Fe K-edge for various samples ($S_0^2=0.74$)

	shell	CN	R(Å)	σ^2	ΔE_0	R factor
Fe foil	Fe-Fe	8	2.47±0.01	0.0049	6.5±1.2	0.0066
	Fe-Fe	6	2.85±0.01	0.0060		
FeMoP-550	Fe-O	6.3±0.2	1.98±0.01	0.0045	-2.4±1.1	0.0039

^aN: coordination numbers; ^bR: bond distance; ^c σ^2 : Debye-Waller factors; ^d ΔE_0 : the inner potential correction. R factor: goodness of fit.

Comments 2: In the line 257, the authors claim that “The Mo K edge XANES spectrum shows that the near edge absorption energy of Fe/SAs@Mo based HNSs is intermediate between that of Mo foil

and MoO₂, demonstrating that the average oxidation state of Mo is between Mo³⁺ and Mo⁴⁺”. However, the average oxidation state of Mo between 0 and 4+ can be concluded. Because there is no reference with Mo³⁺ showed.

Response: Thank the reviewer very much for the critical comments. We are sorry for this misleading expression. We agree that the expression of the valence state of Mo locates in between Mo⁰ and Mo⁴⁺ is more accurate base on Fig. 3d. In response to this comment, the sentence has been revised to: “demonstrating that the average oxidation state of Mo is between Mo⁰ and Mo⁴⁺” (Paragraph 2 on Page 9)

Comments 3: The XPS results show that Mo⁴⁺ and Mo⁶⁺ are the majority. Why the average oxidation state of Mo in XAS is lower than 4+?

Response: The sample detection depth is about several nanometer for XPS, however, it is more than 10 μm for fluorescence-mode XAS, and 1 mm for transmission-mode XAS. The mode used during the XAS test is transmission mode for Mo and is fluorescence mode for Fe, respectively. The thickness of our nanosheet-like catalyst is far less than 10 μm. Therefore, the XAS test detected the entire sample thickness of our catalyst. The surface of catalyst was more easily phosphorised to MoP₂ during the phosphorisation process, thus it is reasonable that the proportion of Mo⁴⁺ and Mo⁶⁺ are the greater than Mo³⁺ in the XPS results.

Comments 4: In the line 293, “they have a smaller slope (35.1 mV dec⁻¹) than those of bulk FeMoP 500 and 20% Pt/C.” However, the Tafel slope of 20% Pt/C is 33.6 mV dec⁻¹ as showed in Figure 4b.

Response: In response to this comment, the sentence has been revised to: “they have a smaller slope (35.6 mV dec⁻¹) than those of bulk FeMoP-500 (89.3 mV dec⁻¹) and other FeMoP-T samples, close to that of 20% Pt/C (36.1 mV dec⁻¹).” (Paragraph 1 on Page 11) The value is slightly different from the original ones as these catalyst were retested in a H₂-saturated electrolyte based on the editor’s suggestion.

Comments 5: Some mistakes should be corrected. In the line 251, “synchrotron radiation based soft X ray absorption near edge structure (XANES)”. It should be hard XANES. In the line 283, “(h) corresponding k₂-weighted FT of EXAFS spectra;” this is different from “FTk₃χ(k)” in the Y axis of Figure 3h. Are they k₂- or k₃-weighted FT of EXAFS?

Response: To address this comment, we have carefully examined the data and confirmed that the expression in the Y axis of Fig. 3h is right and corrected the legend of Fig. 3 (changes highlighted in red in the revised manuscript).

Reviewer #2

General Comments: This manuscript by Lyu et al. reported the fabrication of the single iron atom dispersed Mo-based nanosheet heterostructure developed from a mineral hydrogel. The heterostructured catalysts were applied as efficient hydrogen evolution reaction (HER) electrocatalysts, showing high activity (a small overpotential of 38 mV at 10 mA cm⁻²) and excellent durability (negligible performance decay after 500 h operation). Furthermore, theoretical calculations demonstrated the role of O-coordinated single Fe atoms in the heterostructure. Overall, I think this manuscript is well organized and written. However, several concerns and questions need to be properly addressed before possible consideration for publication in Nature Communications.

Comments 1: The authors claimed this synthesis approach shows “universality”, but I cannot find any evidence in this work. It would be interesting to see how this method can be applied to other single metal dispersed structures.

Response: The strong ionic character of the mineral hydrogel can easily accommodate dispersed metal ionic additives and transition metal salts, which facilitates the manipulation of the targeted single-atoms. To address this comment, other metal ions (e.g., Co²⁺, Ni²⁺, Cu²⁺, Ag⁺, or Mn³⁺, ~ 3 at.% of the sum of Fe and Mo atoms) were added to the Fe³⁺ solution and mixed with the phosphomolybdic acid to prepare different mineral hydrogels. As shown in Fig. R2, the addition of the other ion species did not affect the formation of the mineral hydrogel, indicating the feasibility of the synthesis approach for preparing mineral hydrogels containing other metal ions. We will continue to study these systems in more details in our future work. Fig. R2 is added as Supplementary Fig. 7 on Page 14 in the revised Supplementary Information. The following sentences have been added to the revised manuscript:

“To demonstrate the applicability of the mineral hydrogel for producing other monatomic dispersed catalysts, different ions (e.g., Co²⁺, Ni²⁺, Cu²⁺, Ag⁺, or Mn³⁺, ~ 3 at.% of the sum of Fe and Mo atoms) were added to the Fe³⁺ solution and mixed with the phosphomolybdic acid to prepare different mineral hydrogels. As shown in Supplementary Fig. 7, the addition of the other ions did not affect the formation of the mineral hydrogel.” (Paragraph 1 on Page 6 in revised manuscript)

Fig. R2 Optical photograph of the mineral hydrogels containing added ion species (~ 3 at.% of the sum of Fe and Mo atoms, the molar ratio of Fe³⁺/PMo is 25:1).

Comments 2: If I understand correctly, the authors concluded MoP and MoP₂ as the most active sites for the HER, and the single dispersed Fe atoms enhance the activity of Mo-based catalysts by introducing additional structural vacancies. Here, the single-atom Fe seems to serve as spectators or absorbing *H₂O to trigger the reaction. However, first, if the binding energy of *H₂O on Fe@MoO₂-2 is very small, such weak adsorption cannot compete with other sites showing very strong binding capabilities; second, if Fe is not the active center, why does one need a sufficient amount of Fe to deliver a decent HER activity? Overall, the experiments and calculations do not match well in this work, or the current descriptions have not been well delivered. Possible synergistic effects may need to be further discovered here.

Response: We would like to apologize that our descriptions are misleading due to the wrong expression “MoP and MoP₂ serve as the main locations of activity” on page 20 in the original manuscript. Based on our DFT results, it can be observed that most of the active sites investigated in our catalysts exhibit outstanding H₂O adsorption ability (Fig. 5a), even the Fe@MoO₂-2 sample with weakest H₂O adsorption ability is comparable with Pt(111). According to the computational results of the Gibbs free energy (ΔG_{H^*}) of H* (Fig. 5c), the main active sites for H₂ production are most of the active sites in heterostructured interfacial and monoatomic dispersed models with ΔG_{H^*} in the range of -0.19–0.22 eV except MoP/MoP₂. Moreover, the newly added theoretical investigation on H₂O dissociation ability (Supplementary Fig. 27) reveals that H₂O dissociation is easy to be activated on the surface of our catalyst in which MoP/MoP₂ and Fe@MoO₂-1 perform best. All the findings mentioned above indicate that both the heterostructured interfacial and monoatomic dispersed models acting as effective active sites for HER.

With regard to the HER performance of single dispersed Fe atoms, two representative models named Fe@MoO₂-1 and Fe@MoO₂-2 are investigated. For Fe@MoO₂-1, the single dispersed Fe sites simultaneously exhibit superior properties for H₂O adsorption (-0.92 eV), H₂O dissociation

(thermodynamically downward) and H₂ production (-0.19 eV) (Fig. 5a, c and Supplementary Fig. 27). Thus, single-atom Fe coordinating with one O atom can efficiently act as active center for HER activity. For Fe@MoO₂-2, although the single dispersed Fe sites show the weakest H₂O adsorption ability, it is comparable with Pt(111) and still can absorb H₂O molecules. Its H₂O adsorption efficiency is much lower than the other sites showing very strong binding capabilities, as pointed out by the reviewer. However, the H₂O dissociation at the Fe sites in Fe@MoO₂-2 are thermodynamically downward (Supplementary Fig. 27) and the ΔG_{H^*} has a great value of 0.16 eV. It indicates that the single dispersed Fe atoms coordinating with two O atoms can also act as active center for HER activity, especially contributing to H₂O dissociation and H₂ production. Therefore, the single dispersed Fe atoms in our catalyst play a critical role on the exceptional HER performance. Overall, abundant heterogeneous catalytic sites in our catalyst, both at the heterostructured interfaces and the surface of MoO₂ with single-atom Fe sites, effectively promote the HER activity.

Here we would like to emphasize that the results obtained from experiments and calculations match well in this work. However, the descriptions in the original manuscript have not been well delivered as the reviewer pointed out. Accordingly, the related content leading to misunderstanding is rewritten and more detailed descriptions are revised on Page 20 in the revised manuscript: “(1) The heterostructured interfaces of Fe/SAs@Mo-based-HNSs lead to optimised electronic structures and H* adsorption energies, increasing their intrinsic electrocatalytic activity. (2) The monoatomic dispersed Fe locations contribute to efficient H₂O adsorption and dissociation capability, promoting proton transfer to accelerate the HER performance.”.

Comments 3: Followed by comment 2, the DFT calculations are weak and are hence not reliable to support the experimental conclusions. For example, the electrocatalysts are operated in an alkaline solution, but the DFT calculations are considered on the acidic Volmer step. The calculations can be complicated if one further considers the pH value effect, the solvent effect, and the reaction kinetics of key elementary steps. However, these aspects were not reflected in the current DFT works. The conclusions may be easily altered by different theoretical results.

Response: The reviewer’s comment is very helpful for us to improve our DFT analysis. It is known that one of the most important steps for HER is H₂O dissociation on the surfaces of catalysts especially in alkaline media¹⁻³. Therefore, the H₂O dissociation performance on the surfaces of MoP, MoP₂, MoO₂, MoP/MoP₂, MoP/MoO₂, MoP₂/MoO₂, Fe@MoO₂-1 and Fe@MoO₂-2 are investigated and compared with Pt(111)³. Fig. R3 shows that the H₂O dissociation on the surface sites of MoP₂, MoP/MoO₂ and MoP₂/MoO₂ are thermodynamically upward while the others are thermodynamically downward. Note that although the energy barriers for H₂O dissociation on the surfaces of MoP₂, MoP/MoO₂ and MoP₂/MoO₂ are thermodynamically upward, they are lower than that on the surface

of Pt(111). It indicates that H₂O dissociation is easy to occur on the surface of our catalyst. Specifically, MoP/MoP₂ and Fe@MoO₂-1 shows the greatest H₂O dissociation capability. In addition, it has already been identified that the H₂O adsorption energy for MoP/MoP₂ and Fe@MoO₂-1 are also very excellent. These findings imply that the origin of efficient H₂O adsorption and dissociation capability of our catalyst could mainly be owing to MoP/MoP₂ and Fe@MoO₂-1, and thus promoting faster proton supply to accelerate the HER process.

Fig. R3 is added on Page 28 as Supplementary Fig. 27 in the revised Supplementary Information. The following content is added on Page 15 in the revised manuscript:

“As an important rate-determining factor for HER,¹⁻³ the H₂O dissociation performance on the surfaces of MoP, MoP₂, MoO₂, MoP/MoP₂, MoP/MoO₂, MoP₂/MoO₂, Fe@MoO₂-1 and Fe@MoO₂-2 are further investigated and compared with Pt(111) (Supplementary Fig. 27).³ The H₂O dissociation energies, either thermodynamically upward with low energy barrier or thermodynamically downward, indicate that H₂O dissociation is easy to occur on the surface of catalysts. The results show that the H₂O dissociation on the surface sites of MoP₂, MoP/MoO₂ and MoP₂/MoO₂ are thermodynamically upward, but much lower than that on the surface of Pt(111), while the others, especially MoP/MoP₂ and Fe@MoO₂-1, are thermodynamically downward, indicating the H₂O dissociation is easy to occur on the surface of our catalyst in which MoP/MoP₂ and Fe@MoO₂-1 shows the greatest H₂O dissociation capability. Combined with the excellent H₂O adsorption energy of MoP/MoP₂ and Fe@MoO₂-1 (Fig. 5a), it can be concluded that the origin of efficient H₂O adsorption and dissociation capability of our catalyst owes in part, to MoP/MoP₂ and Fe@MoO₂-1 promoting proton transfer to accelerate the Volmer step of HER.”

Fig. R3 Free energy diagrams of reaction coordinate for water dissociation on the surfaces of single-phase models (MoP, MoP₂ and MoO₂), heterostructured interface models (MoP/MoP₂, MoP/MoO₂ and MoP₂/MoO₂), monoatomic dispersed Fe onto MoO₂ surface models (Fe@MoO₂-1 and Fe@MoO₂-2) and Pt(111)³.

Comments 4: Would other thermochemical treatment temperatures yield single metal dispersed structures?

Response: In this work, we thought the single metal dispersed structures also could be formed when the thermochemical treatment temperature is higher than 500 °C. From the XRD results of the sample treated at ≥ 500 °C (Supplementary Fig. 10 , also shown in the Following Fig. R4), no relate iron oxide or phosphide can be detected in the XRD. The catalyst will aggregate into big and hard products when the thermochemical treatment temperature is higher than 700 °C, and the specific surface area and pore size distribution of the sample treated at 550 °C (denoted as FeMoP-550) is close to that of 500 °C. In addition, we had that there is Fe element in FeMoP-550 with ICP (Table R5). So, we conducted the structural analysis for the sample treated at 550 °C. In addition, from the TEM results (Supplementary Fig. 13), no diffraction rings and lattice fringe relate to iron oxide or phosphide can be found in the SEAD pattern and HRTEM image, Fe element distributed uniformly in the whole materials according to the EDS mapping. To further demonstrate the existing state of Fe, we done the hard X-ray absorption near-edge structure (XANES) for FeMoP-550 at Fe K-edge, the XANES adsorption results and its corresponding fitting results were shown in Fig. R5 and Table R6 (also added as Supplementary Fig. 19 and Supplementary Table 8 in revised Supplementary information). The valence state of Fe in FeMoP-550 is similar to that of Fe/SAs@Mo-based-HNSs, that is between +8/3 and +3. The corresponding Fourier-transform-EXAFS and its wavelet transforms signals (the strong peak at ~ 1.5 Å is attributable to Fe–O bonding, and no Fe–Fe peak is present at 2.47 or 2.85 Å) delineate the Fe atoms are only coordinated with O atoms, thus the Fe atoms are isolated in FeMoP-550. So, in this work, we can get the conclusion that the other thermochemical treatment temperatures can yield single metal dispersed structures. The following sentences have been added to the revised manuscript:

“Although the greater loss of Fe species in a higher pyrolysis temperature (e.g. in 550 °C), the residual Fe atoms were also in the form of monoatomic dispersion (Supplementary Fig. 19, Supplementary Table 8)” (Paragraph 1 on Page 11 in revised manuscript)

Fig. R4 XRD patterns of samples obtained at different phosphorization temperatures.

Table R5. Elemental compositions for FePMoG, FeMoP-450, Fe/SAs@Mo-based-HNSs and FeMoP-550 determined by ICP-OES.

Samples	Fe (mg/kg)	Mo (mg/kg)	P (mg/kg)	O (mg/kg)	$r_{\text{Fe/Mo}}$
FePMoG	146725.83	481927.19	/	/	0.523
FeMoP-450	70586.59	431511.55	/	/	0.281
Fe/SAs@Mo-based HNSs	62367.61	420141.41	/	/	0.255
FeMoP-550	22122.89	447094.81	/	/	0.085

$r_{\text{Fe/Mo}}$ is the atomic ratio of Fe to Mo.

Fig. R5 Spectroscopy of FeMoP-550 at Fe K-edge: (a) Fe K-edge XANES spectra; (b) corresponding k^3 -weighted FT of EXAFS spectra; (c) the corresponding k^3 -weighted FT-EXAFS spectra and fitting line in the R spacing; and (d) wavelet transforms for k^3 -weighted EXAFS signals.

Table R6. EXAFS fitting parameters at the Fe K-edge for various samples ($S_0^2=0.74$)

	shell	CN	R(Å)	σ^2	ΔE_0	R factor
Fe foil	Fe-Fe	8	2.47±0.01	0.0049	-6.5±1.2	0.0066
	Fe-Fe	6	2.85±0.01	0.0060		
FeMoP-550	Fe-O	6.3±0.2	1.98±0.01	0.0045	-2.4±1.1	0.0039

^aN: coordination numbers; ^bR: bond distance; ^c σ^2 : Debye-Waller factors; ^d ΔE_0 : the inner potential correction. R factor: goodness of fit.

Comments 5: How could you know there are no Fe-P bonds but only Fe-O? The coordination environment of the Fe atom is 5.9 O atoms from EXAFS, why do the DFT models only contain 1 or 2 O atoms? This is important because DFT calculation results are very sensitive to the local coordination structure of the metallic centers (see some related references: Small 2022, 18, 2105680; Adv. Mater.

2021, 33, 2103004; Mater. Today Energy 2021, 20, 100653).

Response: The real bond length of Fe-P and Fe-O is about 2.2/2.3 and 1.98 Å, respectively. In the k^3 -weighted FT of EXAFS spectra, for Fe-P, the main peak will locate at the position close to 2.0 Å; for Fe-O, the main peak will appear at the position of ~ 1.5 Å. From the the k^3 -weighted FT of EXAFS spectra (Fig. R6a), it can be observed that the main peak of Fe/SAs@Mo-based-HNSs appear at the location of ~ 1.5 Å. It also can be seen clearly the peak position of the maximum peak lies at ~ 1.5 Å in wavelet transforms for k^3 -weighted EXAFS signals at Fe K-edge (Fig. R6b). Therefore, based on the fitting results and all the analysis from XAFS, we got the conclusion that the Fe atoms is only bonding with O atoms.

We do agree with the reviewer about the importance of local coordination structure on the DFT calculation results as expressed by the sentence “This indicates that the local bonding environment of monoatomic dispersed Fe atoms greatly affects the H₂O adsorption behavior of electrocatalytic sites.” On Page 15 in our original manuscript. The ideal unit cell of MoO₂ shows that Mo atom is coordinated with six O atoms with a bonding distance of ~ 2.0 Å (Fig. R7). Since Fe atoms mainly exist in MoO₂ as substitutional solid solution, it should also be coordinated with six O atoms which is greatly consistent with our EXAFS result that Fe atom is coordinated with 5.9 O atoms. In addition, our EXAFS result for Fe-O bond distance (~ 1.98 Å) is almost equal to Mo-O, implying that replacing Mo by Fe atoms almost does not change the structure of MoO₂. The experimental information indeed indicates that Fe atoms in MoO₂ should be coordinated with about six O atoms. However, it should be noted that the data obtained from EXAFS is the information of average coordination number, which mainly be determined by the coordination number of Fe atoms in MoO₂ interior. Since the Fe active sites contributing to HER are the monoatomic Fe atoms at the surface of the material, their coordination number should be smaller than that in the material interior, existing in the form of unsaturated coordination. Moreover, as mentioned in the section of DFT calculations in Supplementary Information, we have explained “Two different stable monoatomic dispersed Fe onto MoO₂ surface models (bonding with one or two O atoms, named Fe@MoO₂-1 and Fe@MoO₂-2 respectively) are obtained via a series of structure optimization test.” The constructed initial configurations with Fe atoms coordinating more O atoms are unstable. Therefore, we take the DFT models only contain 1 or 2 O coordination atoms as the representative monoatomic Fe models and investigate their HER performance.

We added a brief description in Supplementary Information to elucidate the reason why the monoatomic Fe sites only coordinate with 1 or 2 O atoms in our DFT models which is inconsistent with the result obtained from EXAFS.

The following content is added on Page 5 in the revised Supplementary Information:

“Since the DFT calculation results are very sensitive to the local coordination structure of the

metallic centers⁴⁻⁶, a series of initial configurations with Fe atoms coordinating different number of O atoms are constructed to demonstrate the representative stable models with monoatomic dispersed Fe onto MoO₂. Two different stable monoatomic dispersed Fe onto MoO₂ surface models (bonding with one or two O atoms, named Fe@MoO₂-1 and Fe@MoO₂-2 respectively) are obtained via a series of structure optimization test. The constructed initial configurations with Fe atoms coordinating more O atoms are unstable. It should be noted that the monoatomic Fe sites only coordinate with 1 or 2 O atoms in our DFT models which is much smaller than the result obtained from EXAFS. However, it should be noted that the data obtained from EXAFS is the information of average coordination number, which mainly be determined by the coordination number of Fe atoms in MoO₂ interior. Since the Fe active sites contributing to HER are the monoatomic Fe atoms at the surface of the material, their coordination number should be smaller than that in the material interior, existing in the form of unsaturated coordination. Therefore, the DFT models only containing 1 or 2 O coordination atoms are taken as the representative monoatomic Fe models.”

The papers mentioned by the reviewer are very useful for us to improve the quality of our revised manuscript and are cited in the revised manuscript.

“H₂O adsorption behaviour of electrocatalytic sites⁴⁰⁻⁴².” (Paragraph 1 on Page 14 in revised manuscript) (⁴⁻⁶)

Fig. R6 (a) corresponding k^3 -weighted FT of EXAFS spectra; and (b) wavelet transforms for k^3 -weighted EXAFS signals of Fe/SAs@Mo-based-HNSs at Fe K-edge.

Fig. R7 The unit cell of MoO₂.

Comments 6: In figure 3d, from the statement “The Mo K-edge XANES spectrum shows that the near-edge absorption energy of Fe/SAs@Mo-based-HNSs is intermediate between that of Mo foil and MoO₂, demonstrating that the average oxidation state of Mo is between Mo³⁺ and Mo⁴⁺”, one would agree the valence state of Mo locates in between Mo and Mo⁴⁺, but why in between Mo³⁺ and Mo⁴⁺? The more accurate determinations of valence states of elements are required based on the XANES results.

Response: We are sorry for this misleading of our expression in our original manuscript. We do agree that the expression of the valence state of Mo locates in between Mo⁰ and Mo⁴⁺ is more accurate base on Fig. 3d. At first, we also stated the average oxidation state of Mo is between Mo⁰ and Mo⁴⁺. In consideration of the XRD result, we thought that it was sure the existence form of Mo species is MoO₂, MoP and MoP₂, the valence state of all these Mo species is higher than +3, so we thought it might give a more accurate valence state range of Mo if we stated the valence state of Mo locates in between Mo³⁺ and Mo⁴⁺. Thank you for your correction advisement and such modification “**demonstrating that the average oxidation state of Mo is between Mo⁰ and Mo⁴⁺**” is made in revised manuscript. Based on the advisement of reviewer, we further calculate the valence states of Fe and Mo based on the XANES results. As can be seen in the following Fig. R8, the average oxidation state of Mo and Fe is +3.36 and +3.12, respectively.

Fig. R8 (a) Mo and (b) Fe valence states of Fe/SAs@Mo-based-HNSs calculated from the XANES fitting results.

Comments 7: For the evaporation of Fe species at high pyrolysis temperatures, can you show more examples?

Response: During the whole synthetic procedure, the relative amount of Fe to Mo should be same or close if there is no evaporation of Fe species. We first used XPS to determine the elemental compositions for the original samples and three samples at different treatment temperatures (450, 500 and 550 °C). The value of atomic ratio of Fe/Mo ($r_{Fe/Mo}$) for FePMoG, FeMoP-450, Fe/SAs@Mo-based-HNSs and FeMoP-550 is 0.461, 0.239, 0.222, and 0.058, respectively (Table R7). This value is reduced after phosphorisation, especially this value decreased to a much smaller number at 550 °C, indicating the content of Fe is decrease, that is the Fe species was lose during the phosphorisation procedure. So, we draw the conclusion that the Fe species was evaporated at high pyrolysis temperatures. The TEM EDS results (Table R8) also show similar trends to the XPS results and this result had been added into the revised Supplementary information. To further verify this point, we performed ICP-OES tests on these samples. As can be seen in Table R9, the results also indicate the evaporation of Fe species.

Table R7. Elemental compositions for FePMoG, FeMoP-450, Fe/SAs@Mo-based-HNSs and FeMoP-550 determined by XPS.

Samples	Fe (at%)	Mo (at%)	P (at%)	O (at%)	$r_{Fe/Mo}$
FePMoG	7.8	16.9	0.4	74.9	0.461
FeMoP-450	4.5	18.8	8.1	68.6	0.239
Fe/SAs@Mo-based HNSs	4.6	20.7	10.4	64.3	0.222
FeMoP-550	1.0	17.3	19.8	61.9	0.058

$r_{\text{Fe/Mo}}$ is the atomic ratio of Fe to Mo.

Table R8. Elemental compositions for FePMoG, FeMoP-450, Fe/SAs@Mo-based-HNSs and FeMoP-550 determined by TEM EDS.

Samples	Fe (at%)	Mo (at%)	P (at%)	O (at%)	$r_{\text{Fe/Mo}}$
FePMoG	8.28	15.56	0.51	75.65	0.532
FeMoP-450	5.3	19.7	8.9	66.1	0.269
Fe/SAs@Mo-based HNSs	5.1	21.4	11.3	62.2	0.238
FeMoP-550	1.4	18.2	21.9	58.5	0.077

$r_{\text{Fe/Mo}}$ is the atomic ratio of Fe to Mo.

Table R9. Elemental compositions for FePMoG, FeMoP-450, Fe/SAs@Mo-based-HNSs and FeMoP-550 determined by ICP-OES.

Samples	Fe (mg/kg)	Mo (mg/kg)	P (mg/kg)	O (mg/kg)	$r_{\text{Fe/Mo}}$
FePMoG	146725.83	481927.19	/	/	0.523
FeMoP-450	70586.59	431511.55	/	/	0.281
Fe/SAs@Mo-based HNSs	62367.61	420141.41	/	/	0.255
FeMoP-550	22122.89	447094.81	/	/	0.085

$r_{\text{Fe/Mo}}$ is the atomic ratio of Fe to Mo.

Comments 8: How would the effect of hydrogen spillover be here?

Response: We do agree that it is necessary to elucidate the role of hydrogen spillover for the superior HER performance of our catalyst. In order to investigate the potential effect of hydrogen spillover in our work, the Mo sites on the surface of MoO₂ and MoP/MoP₂, which have good $\Delta E_{\text{H}_2\text{O}}$ and H₂O dissociation ability (Fig. 5a and Supplementary Fig. 27) but poor ΔG_{H^*} (Fig. 5c), are selected as the representative initial sites of hydrogen spillover for the reason of their efficient H* supply while inefficient H₂ production and the redundant H* may migrate to other active sites for H₂ production. On the other hand, the Mo sites in MoP₂/MoO₂ and Fe sites in Fe@MoO₂-1 are selected as the representative sites for H₂ production after hydrogen spillover from MoP/MoP₂ and MoO₂, respectively. Because both of them possess good H₂ production ability (Fig. 5c) and the HER performance could be greatly accelerated if hydrogen spillover can easily occur between these active sites. By the way, we assume that H* may go through MoP₂ when migrating from Mo sites in MoP/MoP₂ to MoP₂/MoO₂ because both of the heterostructured interfaces adjacent to MoP₂. Therefore,

the energy barriers of two hydrogen spillover pathways, $\text{MoP/MoP}_2 \rightarrow \text{MoP}_2 \rightarrow \text{MoP}_2/\text{MoO}_2$ and $\text{MoO}_2 \rightarrow \text{Fe@MoO}_2\text{-1}$, are calculated.

The energy profile for the first hydrogen spillover pathway is shown in Fig. R9. It can be observed that the H^* preferentially adsorbs at Mo site in MoP/MoP_2 with a ΔG_{H^*} value of -0.365 eV (site 1) while showing weak interaction with Mo site in MoP_2 ($\Delta G_{\text{H}^*} = 0.296$ eV, site 2). Such a strong hydrogen capturing at site 1 and weak interaction at site 2 result in a thermodynamic barrier of 0.661 eV. In addition, the transition state (TS) along the migration path from MoP/MoP_2 to MoP_2 shows a kinetic barrier of 0.871 eV. On the other hand, according to the energy profile for the second hydrogen spillover pathway (Fig. R10), there exists a very strong H^* capturing at site 3 (-0.754 eV), leading to a high thermodynamic barrier (0.569 eV) from the Co-Fe bridge site (site 3) to Fe site (site 4) at the surface of $\text{Fe@MoO}_2\text{-1}$. Thus, the spillover process may be severely hindered by these high energy barriers, which implies that hydrogen spillover may play a secondary role on the excellent HER performance of our catalyst. Instead, the overall HER process, H_2O dissociation and H_2 production, may locally carry on at the same active sites.

Fig. R9 Calculated free energy diagram for hydrogen spillover from MoP/MoP_2 to MoP_2 and then to $\text{MoP}_2/\text{MoO}_2$, the insets are the optimized H^* adsorption configurations at various sites along the migration path.

Fig. R10 Calculated free energy diagram for hydrogen spillover from MoO₂ to Fe@MoO₂-1, the insets are the optimized H* adsorption configurations at various sites along the migration path.

Comments 9: Some typos were found in the manuscript, not limited to:

“single atom subtrate precursors” should be “single atom substrate precursors”?

“the common substragte precursors” should be “the common substrate precursors”?

Response: Thank you for your helpful comments. We are sorry for those mistakes. In the revised manuscript, we have carefully corrected the gramma and spells. We have carefully corrected this mistake and other gramma and spells which highlighted in green font in the revised manuscript.

Reviewer #3

General Comments: In this article, the authors have devised a highly efficient HER electrocatalyst composed of porous Fe/SAs@Mo-based-HNSs, which is formed via a novel lowtemperature phosphorisation of environmentally benign and simple self-assembled inorganic–inorganic coordinated FePMoG nanosheets. The author attribute to the good performance of HER to Fe/SAs@Mo-based-HNSs’ optimised electronic structure, enriched interface and boundary phases, large active surface areas and porosities, and the synergetic effect of their single dispersed atoms and heterostructures. However, there are some problems in the article, and the experimental data cannot fully support this result. After making the following major revisions, this manuscript can be published in the *nature communication*.

Comments 1: In Figure 1a, the author makes a quantitative comparison of several substrate precursors.

But there is no evidence to support this comparison. And this figure shows that the mineral hydrogel seems to be a too perfect substrate precursor, because every index is optimal, I think this figure is kind of misleading.

Response: We compared some prominent characters of mineral hydrogel to porous frame works and carbon substrate, each item is further divided into more detailed subdirectories (Table R10, also shown in Supplementary Table 1). The data were got from previous publication and websites. Based on the practical value, complex, toxicity, etc., a score was evaluated (in the range of 0-10). To highlight the advantage of mineral hydrogel, total access score of mineral hydrogel is normalized to 100. Although the score may have certain degree of subjectivity, the details is totally based on the publications, therefore we thought the Fig. 1a can well reveal the merits of these substrate precursors.

Table R10. Comparison of mineral hydrogel, porous framework and carbon substrate used for single atom catalyst production.

		Mineral hydrogel		Porous framework		Carbon	
	content	details	Score ^a	details (e.g. Ref ⁷)	Score ^a	details (e.g. Ref ⁸)	Score ^a
facile fabrication	instrument	centrifuge	10	oven, ice machine, heating agitator, glass reactor, freezer dryer, vacuum pump, filter, sonicator, centrifuge, tube furnace	2	glass beaker, heating agitator, oven, tube furnace, centrifuge, sonicator	3
	procedure	standing, centrifugation,	10	stirring, cooling, heating, vacuum filtration, freeze dry, conjugation, polymerization, sonication	2	stirring, drying, heating, noble metal deposition, sonication, heating and stirring, centrifugation, drying	3
	template	no need	10	no need	10	need release gas to assist the formation of nanosheet	5
	purification	water wash	10	washed with degassed ethanol and diethyl ether, solvothermal	3	water and ethanol wash	5

				wash with organic solvent			
	reaction condition	room temperature	10	ice bath, heated at 120 °C, heated at 300 °C	2	drying at 80 °C, heated at 900 °C, sonication, stirring at 80 °C	2
	total access	simple synthetic procedure, equipment	100	complex synthetic procedure, reaction conditions are complex and numerous	40	complex synthetic procedure, reaction conditions are complex and numerous	30
green synthesis	Materials	Fe(NO ₃) ₃ ·9H ₂ O, phosphomolybdic acid, NaCl, NaH ₂ PO ₂ , ethanol, deionized water	10	CoCl ₂ ·6H ₂ O, chloranilic acid, H ₂ SO ₄ , ethylenediamine, 1-methy-2-pyrrolidinone (NMP), anhydrous NMP, diethyl ether, ethanol, deionized water	1	urea, glucose, Ni(CH ₃ COO) ₂ · 4H ₂ O, ethanol, ethylene glycol, H ₂ PtCl ₆ , ethanol, deionized water, ammonium hydroxide	8
	yield (from raw material to final catalyst)	~80%	8	Synthesis of hexaaminobenzene: ~39.69% conjugation: 50.89% polymerization: 90%	1.5	carbonization: usually <30%	3
	organic release (except ethanol)	none	10	isopropanol, NMP, anhydrous NMP, ethylenediamine, diethyl ether is easy to evaporate	1	Organic waste during the high temperature pyrolysis	1
	solvent	deionized water	10	isopropanol, NMP, ethylenediamine, diethyl ether	1	ethylene glycol and ethanol mixture	1
	Catalyst usage	no	10	Some frame works preparation need catalyst, e.g. covalent organic	3	CVD method to synthesize the carbon need grow the catalyst	4

				framework ⁹		beforehand ¹⁰	
	recycle	few kinds of ions in liquid waste, easy to disposal	10	Some organic solvent can recycle but it is complicate to separate	5	the kind of organic solvent is less and easy to recycle,	
	total access	using common non-toxic inorganic salt, easy recycle	100	using multiple organic solution and some are toxic	10	using common non-toxic inorganic salt, easy recycle	50
time efficiency	precursor synthesis	20 h	8	64 h	3	18 h	8.5
	subsequent synthesis	16.5 h	9	81 h	1	34 h	3
	total access	no equipment needed, just few more cans can produce mineral hydrogel simultaneously	100	some MOF even need several week to prepare and purification ¹¹	20	if use MOF and COF as carbon source, the synthetic time will increase greatly	40
morphology controllability	1D	Ref ¹²	10	Ref ¹³	6	Ref ^{14,15}	3
	2D	This work	10	Ref ^{7,16}	5	Ref ^{8,17}	6
	3D	Ref ¹⁸⁻²⁰	10	Ref ²¹	9	Ref ²²	9
	total access	easy to control the morphology	100	need to change a lot of reagents, reaction condition, even develop a new method	60	need to change a lot of reagents, reaction condition, even develop a new method, involving using the catalyst and using the porous framework to prepare	35
Organic free	raw material	yes	10	no	0	no, but using common organic solvent with low toxicity	5
	products	yes	10	the intermediate products are organic	1	organic waste produced during the high temperature pyrolysis	5
	total	no organic	100	using a variety of	5	the glucose and	50

	access	species during the whole preparation process of mineral hydrogel		and large amount organic solvents and some are toxic and complicate to prepare		ethylene glycol is abundant and non-toxic	
abundance	source	resource of inorganic salts is rich and easy to process	10	the synthesis of some organic solvent is complex and their yield is low	1	noble metal salt is rare and expensive; the organic solvent used in easy to produce	3
	total access	rich in raw materials and easy to prepare	100	using a variety of and large amount organic solvents and some are toxic and complicate to prepare	5	all reagent except the noble metal salt is rich and common; many carbon-based catalyst didn't need to use the noble metal	60
universality	ion species	majority metal ion can be added in the mineral hydrogel	9	Ref ⁷ can use multiple metal ion for conjugation. However, majority need special metal that can coordinate with linker, or absorb limited metal ion ²³⁻²⁶	4	many metal ions can be absorbed in the carbon or carbon precursor, the use of noble metal limit the universality	6
	synthetic method	simple, directly add other metal ion in the precursor solution	10	need to consider the coordinated property of the metal ions, or absorb ability of the framework	3	Need to consider the property of metal ion; the further deposition of noble metal is low efficiency	3
	total access	easy to add other metal ion and have large potential in prepare other metal single atom catalyst	100	can prepare other single atom catalyst but the efficiency and yield is low, synthetic method is complex	30	can prepare other single atom catalyst but the efficiency and yield is low, synthetic method is complex	60

Low cost (USD, per kg produced catalyst)	materials cost	35.39	10	4779.64	2	8045.85	1
	electricity consumption	0.49	10	97.44	1	47.78	4
	total access	only using abundant inorganic salts; simple preparation process; few instruments are used	100	need use many special organic solvent; rare in resources; complex production process; use many different instrument	10	the noble metal salt and some raw material is expensive; not rich in resources; need to use large amount of organic solvent, multiple preparation steps; some method didn't use noble metal and the cost will reduce a lot ²⁷	70

^a These scores are based on the practical value, or complication degree, toxicity (in the range of 0-10); total access score of mineral hydrogel is normalized to 100; the details were refer to the example of 2D porous frame work and carbon.

Comments 2: The synthetic diagram of the material (figure 1b) is too simple, and many important information are not shown. For example, this diagram does not reflect how Fe single atoms are formed, and even the reaction conditions and precursors are not shown.

Response: The reviewer's comment is very helpful for us to improve the quality of the synthetic diagram. To address this comment, we redrew a new synthetic diagram for the whole experimental process. As shown in Fig. R11, the reactants, reaction conditions, precursors, and the reaction process are more clearly presented. The original Fig. 1b have been replaced by Fig. R11 and relate comments to new Fig. 1b have been modified in the revised manuscript. To further illustrate the evolution process of the FePMoG, the original Fig.1b were moved to Supplementary Fig. 7g.

Fig. R11 schematic of synthesis of the Fe/SAs@Mo-based-HNSs electrocatalyst.

Comments 3: Why is Fe SAs only formed by pyrolysis at 500°C, and whether Fe also exists in the form of single atoms at other temperatures? This needs to be confirmed. If Fe SAs can also be formed at other temperatures, the article mentions that these single-atom dispersed heterostructured nanosheets was first developed from a mineral hydrogel, but the performance of bulk FePMo-500 is even better than that of FePMo-450, which may prove that Fe single atoms are not very important for the performance of HER. In addition, from figure s21, the Fe content seems to become very little in the samples after a long time of testing, does this also indicate that Fe SAs is not an active site for HER.

Response: In this work, we didn't declare that the Fe SAs only formed by pyrolysis at 500°C, we thought the single dispersed Fe also exist at other treated temperature. For example, for the FeMoP-550 which was obtained at a pyrolysis temperature of 550°C, no relate iron oxide or phosphide can be detected in the XRD (Supplementary Fig. 10, also shown in the Following Fig. R12), the specific surface area and pore size distribution of FeMoP-550 is close to that of Fe/SAs@Mo-based-HNSs. In addition, the ICP (Table R11) results illustrated that the Fe still existed in FeMoP-550. And from the TEM results (Supplementary Fig. 13), and no diffraction rings and lattice fringe relate to iron oxide or phosphide can be found in the SEAD pattern and HRTEM image, Fe element distributed uniformly in the whole materials according to the EDS mapping. To further demonstrate the existing state of Fe, we done the hard X-ray absorption near-edge structure (XANES) for FeMoP-550 at Fe K-edge, the XANES adsorption results and its corresponding fitting results were shown in Fig. R13 and Table R12 (also added as Supplementary Fig. 19 and Supplementary Table 8). The valence state of Fe in FeMoP-550 is similar to that of Fe/SAs@Mo-based-HNSs, that is between +8/3 and +3. The corresponding Fourier-transform-EXAFS and its wavelet transforms signals (the strong peak at ~1.5 Å is attributable to Fe–O bonding, and no Fe–Fe peak is present at 2.47 or 2.85 Å) delineate the Fe atoms are only coordinated with O atoms, thus the Fe atoms are isolated in FeMoP-550. So, in this work, we can know

that Fe also exists in the form of single atoms at other temperatures.

Fig. R12 XRD patterns of samples obtained at different phosphorization temperatures.

Table R11. Elemental compositions for FePMoG, FeMoP-450, Fe/SAs@Mo-based-HNSs and FeMoP-550 determined by ICP-OES.

Samples	Fe (mg/kg)	Mo (mg/kg)	P (mg/kg)	O (mg/kg)	$r_{\text{Fe/Mo}}$
FePMoG	146725.83	481927.19	/	/	0.523
FeMoP-450	70586.59	431511.55	/	/	0.281
Fe/SAs@Mo- based-HNSs	62367.61	420141.41	/	/	0.255
FeMoP-550	22122.89	447094.81	/	/	0.085

$r_{\text{Fe/Mo}}$ is the atomic ratio of Fe to Mo.

Fig. R13 Spectroscopy of FeMoP-550 at Fe K-edge: (a) Fe K-edge XANES spectra; (b) corresponding k^3 -weighted FT of EXAFS spectra; (c) the corresponding k^3 -weighted FT-EXAFS spectra and fitting line in the R spacing; and (d) wavelet transforms for k^3 -weighted EXAFS signals.

Table R12. EXAFS fitting parameters at the Fe K-edge for various samples ($S_0^2=0.74$)

	shell	CN	R(Å)	σ^2	ΔE_0	R factor
Fe foil	Fe-Fe	8	2.47±0.01	0.0049	-6.5±1.2	0.0066
	Fe-Fe	6	2.85±0.01	0.0060		
FeMoP-550	Fe-O	6.3±0.2	1.98±0.01	0.0045	-2.4±1.1	0.0039

^aN: coordination numbers; ^bR: bond distance; ^c σ^2 : Debye-Waller factors; ^d ΔE_0 : the inner potential correction. R factor: goodness of fit.

The reviewer commented that “but the performance of bulk FeMoP-500 is even better than that of FeMoP-450,” was wrong, as can be seen in Fig. R14, a comparison of the performance between bulk FeMoP-500 and FeMoP-450 is shown, the performance of bulk FeMoP-500 is obviously poorer than that of FeMoP-450. The Fe SAs can also be formed at other temperatures, such as illustration in the

above discussion, the Fe atoms in the FePMo-550 was also exist in the form of monoatomic dispersion. The performance of FePMo-550 is much better than that of bulk FePMo-500, but it performs much inferior to that Fe/SAs@Mo-based-HNSs. The good performance of Fe SAs@Mo-based HNSs is results from the synergetic effect of the optimized phase composition, heterostructured interfaces, and single dispersed atoms. From the DFT simulation results, MoP/MoP₂ and Fe@MoO₂-1 shows the greatest H₂O dissociation capability and the H₂O adsorption energy for MoP/MoP₂ and Fe@MoO₂-1 are also identified to be very excellent. These findings imply that the origin of efficient H₂O adsorption and dissociation capability of our catalyst could mainly be owing to MoP/MoP₂ and Fe@MoO₂-1, and thus promoting faster proton supply to accelerate the HER process. Even though the more Mo phosphides in FePMo-550 will result in more MoP/MoP₂ interfaces, the content of Fe single atoms is decrease (Table R11), so the synergetic effect of the optimized heterostructured interfaces and single dispersed atoms is inadequate and give rise to the inferior performance of FePMo-550. Similar in FePMo-450, the MoP/MoP₂ interfaces and the transform of Fe single atoms are also not enough, therefore FePMo-450 delivers a worse performance compared to Fe/SAs@Mo-based-HNSs. In the Supplementary Fig. 23 (original manuscript is Figure S21), although the contrast of Fe is low, but this is relative, other elements' contrast is also relative low, the contrast can't act as a criteria to judge the content of an element. From Supplementary Table 10 (also shown in the following Table R13), the dissolution of Fe in the electrolyte after stability test is very little, whose concentration is close to the detection limit of ICP-OES equipment. These prove that Fe single atoms play a very important role in electrocatalytic HER process.

Fig. R14 Polarization curves of the bulk FeMoP-500 and FeMoP-450 in 1 M KOH with iR correction.

Table R13. ICP-OES results of the Fe/SAs@Mo-based-HNSs after stability test for 500h at 20 mA cm⁻² current density.

	Fe	Mo	P
Concentration (mmol/L)	0.014	0.009	0.012
Limit of reporting (mmol/L)	0.010	0.0058	0.030

**The lowest concentration of a substance that can be reliably reported by ICP-OE□*

Comments 4: The article said “high HER electrocatalytic activity of Fe/SAs@Mo-based-HNSs is largely attributable to the optimized electronic coupling in their abundant heterostructured active sites, and is supported by the very high electrolyte-accessible surface area of their 2D porous networks”, but from figure s11 and s12, There are also many heterostructured active sites in FeMoP-450 and FeMoP-550, so it is not convincing.

Response: In this work, we conclusion that one of the most important reasons for the superior HER performance of the Fe/SAs@Mo-based-HNSs is the synergetic effect of the optimised phase composition, heterostructured interfaces, and single dispersed atoms leads to optimised electronic structures and H* adsorption energies. The content of each phase and single Fe atoms will lead to different active heterostructured interfaces and single atom sites. From the DFT simulation results, MoP/MoP₂ and Fe@MoO₂-1 shows the greatest H₂O dissociation capability and the H₂O adsorption energy for MoP/MoP₂ and Fe@MoO₂-1 are also identified to be very excellent. These findings imply that the origin of efficient H₂O adsorption and dissociation capability of our catalyst could mainly be owing to MoP/MoP₂ and Fe@MoO₂-1, and thus promoting faster proton supply to accelerate the HER process. Even though the more Mo phosphides in FePMo-550 will result in more MoP/MoP₂ interfaces, the content of Fe single atoms is decrease (Table R10), so the synergetic effect of the optimized heterostructured interfaces and single dispersed atoms is inadequate and give rise to the inferior performance of FePMo-550. Similar in FePMo-450, the MoP/MoP₂ interfaces and the transform of Fe single atoms are also not enough, therefore FePMo-450 delivers a worse performance compared to Fe/SAs@Mo-based-HNSs. Therefore, although there are also many many heterostructured active sites in FeMoP-450 and FeMoP-550, the HER performance of FeMoP-450 and FeMoP-550 is not good enough compared to that of Fe/SAs@Mo-based-HNSs.

Comments 5: There are too many abbreviations in the article. For example, FePMoGs , FePMoG, FePMo and FePMo-T can easily cause confusion in reading.

Response: The reviewer’s comment is very helpful for us to improve the quality of the manuscript. The abbreviation of FePMoGs is replaced with FePMoG, and FePMo is replaced with FePMo-T in

revised manuscript. In addition, as the PMo which is the abbreviation of phosphomolybdic acid in original manuscript is easy to cause confusion with. We have carefully checked the manuscript and corrected these abbreviations in the revision.

References

- 1 Sun, H. et al. Topotactically Transformed Polygonal Mesopores on Ternary Layered Double Hydroxides Exposing Under-Coordinated Metal Centers for Accelerated Water Dissociation. *Adv. Mater.* **32**, 2006784 (2020).
- 2 Zhang, B. et al. Interface engineering: The Ni(OH)₂/MoS₂ heterostructure for highly efficient alkaline hydrogen evolution. *Nano Energy* **37**, 74-80 (2017).
- 3 Li, F. et al. Construction of Porous Mo₃P/Mo Nanobelts as Catalysts for Efficient Water Splitting. *Angew. Chem. Int. Ed.* **57**, 14139-14143 (2018).
- 4 Wu, Z. P. et al. Manipulating the local coordination and electronic structures for efficient electrocatalytic oxygen evolution. *Adv. Mater.* **33**, 2103004 (2021).
- 5 Cheng, X. et al. Engineering local coordination environment of atomically dispersed platinum catalyst via lattice distortion of support for efficient hydrogen evolution reaction. *Mater. Today Energy* **20**, 100653 (2021).
- 6 Tomboc, G. M., Kim, T., Jung, S., Yoon, H. J. & Lee, K. Modulating the Local Coordination Environment of Single-Atom Catalysts for Enhanced Catalytic Performance in Hydrogen/Oxygen Evolution Reaction. *Small* **18**, 2105680 (2022).
- 7 Lin, C. et al. 2D-organic framework confined metal single atoms with the loading reaching the theoretical limit. *Mater. Horiz.* **7**, 2726-2733 (2020).
- 8 Li, P. et al. Nickel single atom-decorated carbon nanosheets as multifunctional electrocatalyst supports toward efficient alkaline hydrogen evolution. *Nano Energy* **83**, 105850 (2021).
- 9 Wu, S. et al. Highly durable organic electrode for sodium-ion batteries via a stabilized α -C radical intermediate. *Nat. Commun.* **7**, 1-11 (2016).
- 10 Bulushev, D. A. et al. Ni-N₄ sites in a single-atom Ni catalyst on N-doped carbon for hydrogen production from formic acid. *J. Catal.* **402**, 264-274 (2021).
- 11 Peng, R.-L. et al. Single-atom implanted two-dimensional MOFs as efficient electrocatalysts for the oxygen evolution reaction. *Inorg. Chem. Front.* **7**, 4661-4668 (2020).
- 12 Li, B. et al. Mineral Hydrogel from Inorganic Salts: Biocompatible Synthesis, All-in-One Charge Storage, and Possible Implications in the Origin of Life. *Adv. Funct. Mater.* **32**, 2109302 (2022).
- 13 Dong, et al. & Huang, Z. One-dimensional amorphous cobalt (II) metal-organic framework nanowire for efficient hydrogen evolution reaction. *Inorg. Chem. Front.* (2022).
- 14 Liu, B. et al. Hybrid heterojunction of molybdenum disulfide/single cobalt atoms anchored nitrogen, sulfur-doped carbon nanotube/cobalt disulfide with multiple active sites for highly efficient hydrogen evolution. *Appl. Catal., B* **298**, 120630 (2021).
- 15 Tavakkoli, M. et al. Electrochemical activation of single-walled carbon nanotubes with pseudo-atomic-scale platinum for the hydrogen evolution reaction. *ACS Catal.* **7**, 3121-3130 (2017).
- 16 Sun, Y. et al. Modulating electronic structure of metal-organic frameworks by introducing atomically dispersed Ru for efficient hydrogen evolution. *Nat. Commun.* **12**, 1-8 (2021).
- 17 Yin, X. P. et al. Engineering the coordination environment of single-atom platinum anchored on graphdiyne for optimizing electrocatalytic hydrogen evolution. *Angew. Chem. Int. Ed.* **57**, 9382-9386 (2018).
- 18 Subrahmanyam, K. S. et al. High-surface-area antimony sulfide chalcogels. *Chem. Mater.* **28**, 7744-7749 (2016).
- 19 Bag, S., Gaudette, A. F., Bussell, M. E. & Kanatzidis, M. G. Spongy chalcogels of non-platinum metals act as

- effective hydrodesulfurization catalysts. *Nat. Chem.* **1**, 217-224 (2009).
- 20 Mondal, C. *et al.* Pure inorganic gel: a new host with tremendous sorption capability. *Chem. Commun.* **49**, 9428-9430 (2013).
- 21 Feng, H. *et al.* Porphyrin-based Ti-MOFs conferred with single-atom Pt for enhanced photocatalytic hydrogen evolution and NO removal. *Chem. Eng. J.* **428**, 132045 (2022).
- 22 Cao, B. *et al.* Tailoring the d-band center of N-doped carbon nanotube arrays with Co₄N nanoparticles and single-atom Co for a superior hydrogen evolution reaction. *NPG Asia Mater.* **13**, 1-14 (2021).
- 23 Jiao, L. & Jiang, H.-L. Metal-organic-framework-based single-atom catalysts for energy applications. *Chem* **5**, 786-804 (2019).
- 24 Dong, P. *et al.* Platinum single atoms anchored on a covalent organic framework: boosting active sites for photocatalytic hydrogen evolution. *ACS Catal.* **11**, 13266-13279 (2021).
- 25 Wang, J., Wang, J., Qi, S. & Zhao, M. Stable multifunctional single-atom catalysts resulting from the synergistic effect of anchored transition-metal atoms and host covalent-organic frameworks. *J. Phys. Chem. C* **124**, 17675-17683 (2020).
- 26 Ji, Y., Dong, H., Liu, C. & Li, Y. Two-dimensional π -conjugated metal-organic nanosheets as single-atom catalysts for the hydrogen evolution reaction. *Nanoscale* **11**, 454-458 (2019).
- 27 Zhu, Z. *et al.* Coexisting single-atomic Fe and Ni sites on hierarchically ordered porous carbon as a highly efficient ORR electrocatalyst. *Adv. Mater.* **32**, 2004670 (2020).

REVIEWERS' COMMENTS

Reviewer #1 (Remarks to the Author):

The response to our comments is good. All concerns are addressed. I would like to recommend accepting this manuscript.

Reviewer #2 (Remarks to the Author):

In revision, the authors have carefully addressed most of the comments and improved the manuscript significantly. Although some answers may lead to some further questions, this would be interesting discussions in the field.

Few minor issues:

The resolution of the figures is low.

How do the authors determine the valence state of elements by the positions of the adsorption edge? Is it in a linear correlation or else? Please provide details.

Line 232, is it safe to state "Fe is only coordinated with O"? Maybe "mostly" is better here.

I recommend publication in Nature Communications after addressing the above minor comments.

Responses to the reviewers' comments

Ms. Ref. No.: NCOMMS-22-13743A

Title: "2D mineral hydrogel-derived single atoms-anchored heterostructures for ultrastable hydrogen evolution "

First of all, we would like to thank the reviewers for their recognition of our revisions and responses. Below, we list the reviewers' remaining concerns in **blue** text and our responses to each in **black** text. We have denoted our updates in the revised manuscript using **red** text for easy identification. We have adopted every reviewer suggestion and we are now confident that our manuscript is suitable for publication in *Nature Communications*.

Reviewer #1

General Comments: The response to our comments is good. All concerns are addressed. I would like to recommend accepting this manuscript.

Response: We are pleased that the reviewer was satisfied with our response. We thank again for his constructive comments in improving our quality of the manuscript.

Reviewer #2

General Comments: In revision, the authors have carefully addressed most of the comments and improved the manuscript significantly. Although some answers may lead to some further questions, this would be interesting discussions in the field.

Few minor issues:

Comments 1: The resolution of the figures is low.

Response: To address this comment, the resolution of figures is increased and re-inserted to the manuscript in the final version. The resolution may be reduced during the conversion to PDF file for reviewing. In final version submission, figures as individual vector files in the main article are provided.

Comments 2: How do the authors determine the valence state of elements by the positions of the adsorption edge? Is it in a linear correlation or else? Please provide details.

Response: In XAFS, valence is judged by the near edge part (XANES), and there are different judgment methods according to the different tested side bands. There are two common test edges, K-

edge and L3-edge. For K-edge, the change of its valence state is judged by the change of the position of the absorption edge, and the valence state increases when it shifts to the high energy. For L3- edge, the change of valence state is judged by the intensity of the white line peak. The higher the white line peak, the higher the valence state. Usually, the valence is corresponding to the first peak of the first derivative of the XAFS curve, which the energy position of this peak can be directly read after simple processing by using the data fitting processing software of XAFS. Theoretically, the valence state is positive linear relationship to absorption edge energy, linear equations were established by testing standard samples with different valence states, then the valence state can be obtained by substituting the absorption edge data of the peak into the equation. In fact, the deviation of this value based on XAFS from the real valence state is quite large. So XAFS usually cannot be used to determine the exact valence value but more for the change of the valence, bonding element and bonding length. Therefore, we didn't claim the specific valence value in the revised manuscript.

Comments 3: Line 232, is it safe to state "Fe is only coordinated with O"? Maybe "mostly" is better here.

Response: The reviewer's suggestion is very helpful for us to draw more accurate conclusions. The following sentences have been revised in the revised manuscript:

“**is mostly coordinated with O**” (Paragraph 4 on Page 7 in revised manuscript)

I recommend publication in Nature Communications after addressing the above minor comments.

Response: We have adopted every reviewer suggestion and revised the manuscript carefully to address above minor comments. Once again, we would like to thank the reviewer for their hard work.